



# Tracking city CO$_2$ emissions from space using a high resolution inverse modeling approach: A case study for Berlin, Germany

Dhanyalekshmi Pillai[1,2*], Michael Buchwitz[1], Christoph Gerbig[2], Thomas Koch[2], Maximilian Reuter[1],Heinrich Bovensmann[1], Julia Marshall[2], and John P. Burrows[1]

[1] Institute of Environmental Physics, University of Bremen, Bremen, Germany
[2] Max Planck Institute for Biogeochemistry, Jena, Germany
[2*] now at: Max Planck Institute for Biogeochemistry, Jena, Germany

*Correspondence to*: Dhanyalekshmi Pillai (kdhanya@bgc-jena.mpg.de)

**Abstract.** Currently 52% of the world's population resides in urban areas and as a consequence, approximately 70% of fossil fuel emissions of CO$_2$ arise from cities. This fact in combination with large uncertainties associated with quantifying urban emissions due to lack of appropriate measurements makes it crucial to obtain new measurements useful to identify and quantify urban emissions. This is required, for example, for the assessment of emission mitigation strategies and their effectiveness. Here we investigate the potential of a satellite mission like Carbon Monitoring Satellite (CarbonSat), proposed to the European Space Agency (ESA) − to retrieve the city emissions globally, taking into account a realistic description of the expected retrieval errors, the spatiotemporal distribution of CO$_2$ fluxes, and atmospheric transport. To achieve this we use (i) a high-resolution modeling framework consisting of the Weather Research Forecasting model with a greenhouse gas module (WRF-GHG), which is used to simulate the atmospheric observations of column averaged CO$_2$ dry air mole fractions (XCO$_2$), and (ii) a Bayesian inversion method to derive anthropogenic CO$_2$ emissions and their errors from the CarbonSat XCO$_2$ observations. We focus our analysis on Berlin in Germany using CarbonSat's cloud-free overpasses for one reference year. The dense (wide swath) CarbonSat simulated observations with high-spatial resolution (approx. 2 km x 2 km) permits one to map the city CO$_2$ emission plume with a peak enhancement of typically 0.8-1.35 ppm relative to the background. By performing a Bayesian inversion, it is shown that the random error (RE) of the retrieved Berlin CO$_2$ emission for a single overpass is typically less than 8 to 10 MtCO$_2$ yr$^{-1}$ (about 15 to 20% of the total city emission). The range of systematic errors (SE) of the retrieved fluxes due to various sources of error (measurement, modeling, and inventories) is also quantified. Depending on the assumptions made, the SE is less than about 6 to 10 MtCO$_2$ yr$^{-1}$ for most cases. We find that in particular systematic modeling-related errors can be quite high during the summer months due to substantial XCO$_2$ variations caused by biogenic CO$_2$ fluxes at and around the target region. When making the extreme worst-case assumption that biospheric XCO$_2$ variations cannot be modeled at all (which is overly pessimistic), the SE of the retrieved emission is found to be larger than 10 MtCO$_2$ yr$^{-1}$ for about half of the sufficiently cloud-free overpasses, and for some of the overpasses we found that SE may even be on the order of magnitude of the anthropogenic emission. This indicates that biogenic XCO$_2$ variations cannot be neglected but must be considered during forward and/or inverse modeling. Overall, we conclude that CarbonSat is well suited to obtain city-scale CO$_2$ emissions as needed to enhance our current understanding of anthropogenic carbon fluxes and that CarbonSat or CarbonSat-like satellites should be an important component of a future global carbon emission monitoring system.

## 1. Introduction

One of the main objectives of any climate policy initiative is to limit atmospheric greenhouse gas emissions resulting from anthropogenic activity to a level that minimizes adverse modification of the climate system. An essential component in attaining this goal is the accurate quantification of emissions at national and state levels in order to independently verify the implemented climate change mitigation and adaptation measures. In the context of CO$_2$, cities are significant contributors of emissions, giving rise to approximately 70 % of the total anthropogenic



emissions (Canadell et al., 2010). However, there exist large uncertainties associated with quantifying urban emissions. This makes it difficult to assess the efficacy of any emission management schemes at urban scales.

While mitigation efforts are being taken in some cities around the globe, they lack objective, observation-based methods to verify their outcomes (Pacala et al., 2010). Some observation-based attempts have been made with a

focus on deriving city-scale emissions in a variety of urban environments (Bergeron and Strachan, 2011; Levin et al., 2011; Mays et al., 2009; Wang et al., 2010; Zimnoch et al., 2010). However, none of these approaches is able to account for $CO_2$ emissions from urban areas with the accuracy required for verification, nor are they easily adaptable to other locations. As a result our current emission estimates are purely based on inventories (bottom-up approach), which have large uncertainties due to many unresolved processes related to spatial and temporal

heterogeneity of emission fluxes and local transport phenomena (Amstel et al., 1999; Gregg et al., 2008; Marland, 2008; White et al., 2011). Recent revelations about the inaccuracy of the knowledge of motor vehicle emissions emphasize this point.

As uncertainties, otherwise known as the sum of systematic and stochastic error, usually increase with increasing spatial resolution, emission estimates are not often available at a scale relevant for urban emissions (Oda and

Maksyutov, 2011). This is problematic in terms of judging the effectiveness of emission reduction schemes or designing new management strategies for emission trading. Furthermore, uncertainties in emission estimates impose important limitations on regional carbon budget estimations derived by most atmospheric inverse frameworks (top-down approach), in which anthropogenic emission fluxes are assumed to be well-known (Corbin et al., 2010; Göckede et al., 2010; Gurney et al., 2002, 2005).

The key limitations to constrain emission fluxes at urban scales via inverse modeling are the unavailability of direct, continuous and high frequency atmospheric $CO_2$ measurements representing $CO_2$ enhancement in urban domains, as well as the inability of current inverse modeling systems to capture the fine-scale variability caused by the atmospheric transport and emission processes at a scale relevant for urban emissions (e.g. Bréon et al., 2015). An assessment study based on ground-based measurements indicated potential drawbacks of using $CO_2$ surface

measurements for emission verification, and strongly recommended the use of sufficiently accurate column averaged $CO_2$ dry air mole fractions, denoted as $XCO_2$, measured from the ground and/or space as the best approach to detect and quantify emissions and emission trends from urban regions (McKain et al., 2012). An effective observation-based scheme is able to disentangle anthropogenic emissions from $CO_2$ fluxes originating from biosphere-atmosphere exchange.

Despite its importance, none of the existing satellites has been specifically designed and focused on observing $XCO_2$ at urban scales. However the first attempt to detect and quantify anthropogenic urban area $CO_2$ emissions from space was initiated with the launch of SCIAMACHY onboard ENVISAT (2002-2012) (Burrows et al., 1995; Bovensmann et al., 1999), which had a variety of atmospheric trace gas targets and applications. This has been followed by TANSO onboard GOSAT (launched in 2009) (Kuze et al., 2009)

Analysis of SCIAMACHY $XCO_2$ retrievals revealed that regionally elevated atmospheric $XCO_2$ over highly populated regions correlates well with anthropogenic $CO_2$ emissions in terms of relative emission increase per year (Schneising et al., 2008, 2013). However, these analyses are limited to large and intense emission regions owing to the coarse spatial resolution (~ 60 km x 30 km) of the SCIAMACHY measurements. Reuter et al. (2014) also presents results related to anthropogenic $CO_2$ emissions for large areas using an assessment of SCIAMACHY $XCO_2$

and $NO_2$ retrievals.

By using GOSAT observations, Kort et al. (2013) reported significant enhancements of $XCO_2$ over megacities ($3.2\pm1.5$ ppm for Los Angeles and $2.4\pm1.2$ ppm for Mumbai), and argued that these enhancements can be exploited to track anthropogenic emission trends over megacities. However, constraining fossil fuel $CO_2$ emissions by using GOSAT $XCO_2$ retrievals is limited by the sparseness of the GOSAT data (Keppel-Aleks et al., 2013). Another





satellite mission, OCO-2, has been launched in 2014, with the aim of measuring global $XCO_2$ with the precision, resolution, and coverage needed to characterize $CO_2$ sources and sinks at regional scales ($\geq$ 1000km) (Crisp et al., 2004). In additional to these, there have been some recent attempts to utilize ground-based measurements of $XCO_2$ to constrain emissions from cities such as Los Angeles (Wong et al., 2015) and Berlin (Hase et al., 2015).

In an effort to overcome these limitations and to achieve $XCO_2$ observations with the precision and accuracy, spatiotemporal coverage, resolution, and sensitivity to near-surface concentration variations that are required to derive emissions at urban scales, a satellite mission has been proposed to the European Space Agency (ESA): Carbon Monitoring Satellite (CarbonSat) (Bovensmann et al., 2010). CarbonSat aims to measure $XCO_2$ and $XCH_4$ at a high spatial resolution (approx. 2 km × 2 km), with good spatial coverage via continuous imaging across a wide
swath. The goal swath width is 500 km, but a smaller swath width will likely be implemented to limit cost (ESA, 2015).

In this study, we investigated two potential measurement swath widths: 500 km (goal requirement) and 240 km (breakthrough requirement). As a result of its relatively wide swath and high spatial resolution, CarbonSat is designed to disentangle natural and anthropogenic sources of $CO_2$ and $CH_4$ from localized sources such as cities,
power plants, methane seeps, and landfills, by utilizing its unique greenhouse gas imaging capability achieved by its high spatiotemporal coverage and resolution. More details on the mission and the current instrument concept are given in Buchwitz et al. (2013a) and in ESA, 2015.

The goal of the present study is to assess an instrument like CarbonSat's capability to quantify emission patterns of moderate to strong localized sources, taking into account a realistic description of the retrieval errors as given in
Buchwitz et al. (2013a), the spatiotemporal distributions of $CO_2$ emissions, and atmospheric transport. Here we present results focusing on Berlin in Germany, being a large city, but not a megacity. According to the classification of Globalization and World Cities (GaWC) for the year 2012 (http://www.lboro.ac.uk/gawc/gawcworlds.html), Berlin is categorized as a "Beta level" city that provides a moderate economic contribution to the world economy. Berlin is located in the northeast of Germany (see Fig. 1) and is relatively isolated, i.e. it is not a part of a large
agglomeration of several cities. This permits us to clearly identify the anthropogenic $CO_2$ emission plume of Berlin from a single CarbonSat "$XCO_2$ image". We use a high-resolution modeling framework, comprising the Weather Research Forecasting (WRF) model combined with a greenhouse gas module (WRF-GHG, Beck et al., 2011) and the Vegetation Photosynthesis Respiration Model (VPRM) to simulate $CO_2$ mixing ratios for a domain centered on Berlin. An analysis is carried out for CarbonSat's cloud-free overpasses for one reference year by applying a simple
Bayesian inversion scheme to estimate the emission budget with associated uncertainty. A preliminary analysis using a least-squares-fitting algorithm was reported in Buchwitz et al. (2013b), but here we present more detailed analysis, which differs from the previous study as follows: the present study (1) uses high-resolution model simulations for each cloud-free CarbonSat overpass over Berlin for the simulated year 2008, (2) prescribes the updated emission inventory including hourly variations, (3)  utilizes a Bayesian inversion approach, and (4)
examines more scenarios to extend the error analysis study.

## 2.  WRF-GHG inverse modeling system

A high-resolution inverse modeling system, utilizing atmospheric $XCO_2$ measurements at high spatial and temporal resolution, is used to retrieve the $CO_2$ emissions at an urban scale. It comprises two components: the WRF-GHG model linking atmospheric transport and the fluxes to realistically represent the distribution of atmospheric $CO_2$
mixing ratios, and a Bayesian inversion technique to optimize the fluxes. One primary objective is to quantify the uncertainties in the retrieved anthropogenic $CO_2$ emission fluxes resulting from typical and reasonable estimates of the systematic and random error of the $XCO_2$ measurements for an instrument like CarbonSat for the spatial resolution of 2km x 2km and the uncertainty in *a priori* knowledge of the surface flux of $CO_2$. For this, we used WRF-GHG forward simulations as the "true" representation of the atmospheric $CO_2$ concentrations and the
associated fluxes as the "true fluxes" to be retrieved.  Hence the deviation in the retrieved fluxes (via inverse





optimization) relative to the "true fluxes" is caused by the CarbonSat simulated observation errors and the modeling errors (including the use of different emission inventories) depending on different scenarios analyzed. Each component of the inverse modeling system is described in the following.

## 2.1.  WRF-GHG forward model simulations

The present study uses the WRF-GHG (version WRFv3.4) forward simulations of $CO_2$ concentrations at high spatial (10 km × 10 km) and temporal (1 hour) resolutions for all of CarbonSat's overpasses over Berlin in the year 2008. The WRF-GHG modeling system has already been used in several regional studies and has shown remarkable performance in capturing fine-scale spatial variability of $CO_2$ mixing ratios (e.g. Ahmadov et al., 2007, 2009; Pillai et al., 2010, 2011, 2012). The model domain describes a region (spatial extent of ~ 900 km × 900 km) centered over

Berlin (Fig.1) and the simulations use 41 vertical levels (the thickness of the lowest layer is about 18 m). Simulations are conducted separately for each day for a period of 30 hours, including a meteorological spin-up time of 6 hours starting at 18 UTC the previous day.

The initial and lateral boundary conditions of the meteorological variables, the sea surface temperature (SST) and the soil initialization fields for each run are prescribed from the European Centre for Medium-Range Weather

Forecasts (ECMWF) model analysis data (http://www.ecmwf.int) with a spatial resolution of about 25 km and 6-hourly temporal intervals. As initial atmospheric $CO_2$ fields and the lateral boundary concentrations, simulations use global $CO_2$ concentration simulations by the atmospheric Tracer transport Model 3 (TM3) with a spatial resolution of 4° × 5°, 19 vertical levels and a temporal resolution of 3 hours (Heimann and Körner, 2003). TM3 simulations used for this study are generated by a forward transport simulation of fluxes that have been optimized using a global

network of $CO_2$ observing stations (Rödenbeck, 2005). Biospheric fluxes within the regional domain are calculated online in WRF-GHG with a diagnostic biospheric model, the Vegetation and Photosynthesis and Respiration Model (VPRM), utilizing remote sensing products and meteorological data at high temporal and spatial resolutions (Mahadevan et al., 2008). To obtain more realistic estimates of biospheric fluxes, a set of parameters in the VPRM, specific for each vegetation class, have been optimized against eddy flux observations obtained during the

CarboEurope IP experiment at various sites (21 measurements sites) under different vegetation types within Europe (Pillai et al., 2012). An overview of the flux optimization is shown in Fig. 2. Regional oceanic fluxes are neglected here since their contribution is insignificant in the context of the present study.

### 2.1.1. Fossil fuel emission fluxes

The anthropogenic $CO_2$ emission fluxes are based on the EDGAR (Emission Database for Global Atmospheric Research, version 4.1, year 2008) global inventory with a spatial resolution of 0.1° x 0.1°. EDGAR is an annually varying database, but we apply time factors in order to provide hourly emissions. The time factors for seasonal, daily, and diurnal variations are based on the step-function time profiles published on the former EDGAR website: http://themasites.pbl.nl/images/temporal-variation-TROTREP_POET_doc_v2_tcm61-47632.xls (see Kretschmer et

al. (2014); Steinbach et al. (2011) for further details). WRF-GHG simulations using these EDGAR emissions are treated as the real distribution of atmospheric $CO_2$ (hereafter referred to as "true $CO_2$ conc."), and the associated EDGAR fluxes as "true fluxes".

In order to examine the impact of the spatio-temporal distribution of fossil fuel emission structures on atmospheric $CO_2$ and to quantify the associated uncertainties in the optimized fluxes, we use different emission data as the prior

emissions, namely those compiled by the Institut für Energiewirtschaft und Rationelle Energieanwendung (IER inventory), University of Stuttgart, (http://carboeurope.ier.uni-stuttgart.de) for the year 2000, at a spatio-temporal resolutions of 10 km and 1 hour. Temporal variations in the IER inventory include traffic rush hours, difference in power demand between weekdays and weekends, domestic heating, and air conditioning (Pregger and Friedrich, 2007). While utilizing the IER year 2000 database to represent the simulation year (2008), we apply scaling factors

in a manner similar to that in Pillai et al. (2011) to preserve the temporal emission pattern differences between





weekdays and weekends. Simulations using the IER database are used as the current knowledge about the atmospheric concentration for the inverse optimization described in Sec. 4.3.

Both these emission fluxes are re-gridded to WRF-GHG's 10 km Lambert Conformal Conic projection grid, conserving the total mass of emissions. These hourly fluxes are added separately to the first model layer, and
transported separately as tagged tracers. Figure 3 shows a spatial map of the averaged EDGAR and IER emission fluxes over all the cloud-free overpasses at a certain hour as well as their differences for the model domain. Strong emissions associated with large industrial areas and cities can be seen well in both inventories. In general, both emission inventories show good consistency in terms of spatial emission structures; however significant differences in emission intensities (magnitude) between the inventories, especially for large cities and power plants, are
common (Fig. 3c). These differences are larger for emissions resulting from power plants than for those from cities. Figure 4 shows the temporal variability of urban-scale emission fluxes in hourly, weekly and monthly averaged time scales for a region around Berlin (~ 100 km × 100 km). For Berlin emissions, considerable differences in temporal variations are found between both inventories, with maximum values of 22.5, 18.5, and 24.0 MtCO$_2$yr$^{-1}$ for hourly, weekly and monthly averaged timescales respectively. As compared to the IER inventory, the EDGAR inventory
shows consistently larger emissions for Berlin. The seasonal variability exhibited by EDGAR Berlin emissions is substantially larger than that of the IER inventory. Larger emissions are seen in the EDGAR inventory in winter months, with values approximately a factor of 1.5 higher than those in summer months. This results from the increased demand of domestic heating in winter. In terms of the seasonal variability of the Berlin city emissions, the IER inventory shows a relatively small difference in winter-summer emission patterns (temporal) as compared to
EDGAR, and shows overall larger emissions in winter. Both inventories show lower emissions during weekends, consistent with the reduced demand of transportation and power consumption. The hourly averaged Berlin emissions provided by both inventories display peak values during 7 to 9 am and 5 to 7 pm (local times), reflecting morning and evening rush hours in terms of city traffic. Interestingly the IER Berlin emissions show "delayed" morning rush hours on weekends, with a maximum value around 11 am (local time).

The significant difference between these inventories in both temporal and spatial scales implies that our current knowledge of urban-scale emissions is inadequate, even for Central Europe, which is relatively well characterized in terms of emissions compared to many other parts of the world. Note that a part of these emission differences is likely due to the different data compilation years of the IER and EDGAR inventories. This "knowledge gap" is also important in inverse-modeling-based estimations of the source-sink distribution of CO$_2$, in which fossil fuel fluxes
are generally assumed to be known. How critical the effect of this assumption is depends on the impact of these differences in emissions (emission uncertainties) on modeled atmospheric mixing ratios, as well as on the transport errors that are included in the model-data mismatch error in the inverse modeling framework. The impact of emission uncertainties is further discussed in Sec. 4.1.

## 2.2. Inverse optimization technique

The inverse optimization utilizes observational constraints to adjust a subset of parameters $\boldsymbol{\lambda}$ out of model parameters $\boldsymbol{p}$ in the surface flux model $\boldsymbol{f_m(p)}$ in order to obtain a modeled concentration consistent with the observations. Hence the anthropogenic atmospheric concentration $\boldsymbol{c}$ (column averaged dry air mole fraction) at different locations and times can be represented as:

$$\boldsymbol{c} - \boldsymbol{c}_{bg} = \mathbf{F}\,\boldsymbol{f_m}(\boldsymbol{\lambda}) + \boldsymbol{\varepsilon}_{error} \qquad (1)$$

Here, the matrix $\mathbf{F}$ links the atmospheric concentration to a vector $\boldsymbol{f_m}(\boldsymbol{\lambda})$ whose dimension is equal to the total number of surface flux elements, multiplied by total time steps. The vector $\boldsymbol{c}_{bg}$ is the background column averaged dry air mole fraction i.e. the concentration due to the advection of upstream tracer concentrations. For the inversion, $\boldsymbol{f_m}(\boldsymbol{\lambda})$ is assumed to be linearly dependent on $\boldsymbol{\lambda}$ and is expressed as:





$$f_m(\lambda) = \boldsymbol{\phi}\,\lambda \tag{2}$$

where $\lambda$ represents a vector of daily scaling factors of surface fluxes, and $\boldsymbol{\phi}$ represents the surface flux field over the model domain.

A linear model is obtained by combining Eq(s). 1 and 2:

$$y = \mathbf{K}\,\lambda + \boldsymbol{\varepsilon}_{error} \tag{3}$$

where the measurement vector $y$ is given by

$$y = c - c_{bg} \tag{4}$$

and, $c_{bg}$ is obtained by linearizing the model with a reference state $\lambda_0 = 0$ (see Eq. 1).

The Jacobian matrix that represents the sensitivity of the observations $y$ to the state vector $\lambda$ is given by

$$\mathbf{K} = \mathbf{F}\,\boldsymbol{\phi} \tag{5}$$

The state vector and the Jacobian matrix are further described in Sec. 3.2. *A priori* knowledge of the surface fluxes, $\lambda_{prior}$, along with their uncertainties is incorporated in the Bayesian formulation. The term, $\boldsymbol{\varepsilon}_{error}$, is assumed to follow the Gaussian distribution described by the error covariance matrices of the measurements, $\mathbf{S}_e$ and the prior estimate, $\mathbf{S}_{prior}$. The posterior estimate of $\lambda$ is obtained by minimizing the cost function, $J$, which is given as:

$$J(\lambda) = (y - \mathbf{K}\,\lambda)^T \mathbf{S}_e^{-1} (y - \mathbf{K}\,\lambda) + (\lambda - \lambda_{prior})^T \mathbf{S}_{prior}^{-1}(\lambda - \lambda_{prior}) \tag{6}$$

Analytically solving for the minimum of Eq. (4) gives the optimal estimate of the state vector of the scaling factors $\hat{\lambda}$, as well as the associated error covariance matrix of $\hat{\lambda}$, termed as the posterior uncertainty, $\mathbf{S}_{\hat{\lambda}}$. These are expressed as follows (Rodgers, 2000):

$$\hat{\lambda} = (\mathbf{K}^T \mathbf{S}_e^{-1} \mathbf{K} + \mathbf{S}_{prior}^{-1})^{-1}(\mathbf{K}^T \mathbf{S}_e^{-1} y + \mathbf{S}_{prior}^{-1}\lambda_{prior}) \tag{7}$$

$$\mathbf{S}_{\hat{\lambda}} = (\mathbf{K}^T \mathbf{S}_e^{-1} \mathbf{K} + \mathbf{S}_{prior}^{-1})^{-1} \tag{8}$$

## 3. Bayesian Inversion of CarbonSat measurements

### 3.1. Pseudo observations

The inversion utilizes a one year dataset of CarbonSat simulated observations at a spatial resolution of 2 km × 2 km, generated using the WRF-GHG forward model (10 km × 10 km) as described in Sect. 2.1 and CarbonSat's retrieval error (2 km × 2 km), estimated using an error parameterization scheme based on the measurement characteristics as described in Buchwitz et al. (2013a). The error parameterization scheme, described in detail in Buchwitz et al.(2013a), is based on six parameters consisting of solar zenith angle (SZA) and scattering-related parameters such as albedo in the near-infrared (NIR) and the first shortwave-infrared (SWIR-1) bands, cirrus optical depth (COD), cirrus top height (CTH), and aerosol optical depth (AOD) at 550 nm. We use the "Level 2 error dataset" (L2e files), described in Buchwitz et al. (2013a), that contains the random and systematic errors of CarbonSat's $XCO_2$ retrievals based on the error parameterization scheme. CarbonSat is assumed to follow an orbit similar to NASA's Terra satellite (www.nasa.gov/terra/), but with an equator crossing time of 11:30 a.m. Hence, for specifying the CarbonSat's geolocation, the L2e files utilize the geolocation provided in the Terra Level 1 dataset for the year 2008, but modified to consider the difference in equator crossing time. This dataset contains fields such as geodetic coordinates, ground elevation, and solar and satellite zenith angles etc., determined using the spacecraft attitude and orbit, a digital elevation model, and information derived from various other datasets such as the Filled Land Surface Albedo Product, generated from MOD43B3 (http://modis-atmos.gsfc.nasa.gov/ALBEDO/) at a spatial resolution of 1 minute (2 km at equator, and < 1 km at the poles), which is used to account for surface albedo. The cirrus





parameters are represented using a spatiotemporally smoothed ($8^o \times 8^o$ and 3 months) dataset of COD and CTH, originally derived from CALIOP (Cloud-Aerosol LIdar with Orthogonal Polarization) onboard CALIPSO (Cloud-Aerosol Lidar and Infrared Pathfinder Satellite Observations, Winker et al., 2009). Global aerosol data products from the "GEMS project" (http://gems.ecmwf.int/) at a spatiotemporal resolution of $1.125^o \times 1.125^o$ and 12 hourly are used to account for aerosols (AOD). This dataset is based on the assimilation of MODIS data and we use the AOD at 550 nm. As described in Buchwitz et al. (2013a), the L2e dataset only contains those Carbonsat simulated observations which are approximately cloud-free as determined using a cloud mask obtained from MODIS Terra (using the MODIS cloud cover data product (MOD35) at a spatial resolution of about 1 km × 1 km). As the remaining ground pixels may still suffer from cloud contamination (e.g., due to "too high" amounts of thin cirrus) or other disturbances, a quality filtering scheme is applied which is based on retrieved (e.g., COD and AOD) and known quantities (e.g., SZA). The quality filtering scheme is described in Buchwitz et al. (2013a) and we use here only those ground pixels which are considered "good" according to this scheme.

Initially, we have identified all the potentially useful Berlin overpasses, i.e., overpasses where at least some CarbonSat simulated observations are present over Berlin and surroundings for a given CarbonSat orbit. We found that the maximum number of observations is obtained during the summer months due to most favorable observation conditions (less clouds for extended time periods and regions, high SZA, etc.). In total, there are 41 days (orbits) of potentially useful overpasses over Berlin for the year 2008 for a swath width of 500 km. Note that the number of overpasses are smaller in the figures shown later. This is because of an additional quality filtering procedure applied after the inverse optimization that is based on retrieved random errors, as explained later.

### 3.2. Definition of the state vector and Jacobian matrix

In the present study, the state vector $\boldsymbol{\lambda}$ (the scalable parameter of the emission flux) corresponds to the scaling factor of emission fluxes for a trimmed model domain, i.e., a region around Berlin (spatial extent: approximately 100 km × 100 km, hereafter referred to as the "target region" (TR). The temporal resolution of $\boldsymbol{\lambda}$ is set to be daily, assuming no spatial variations within the target region. The prior value of this scaling factor, $\boldsymbol{\lambda}_{prior}$, is set to unity.

The Jacobian matrix $\mathbf{K}$ relates the measurement vector $\boldsymbol{y}$ to the state vector $\boldsymbol{\lambda}$, and has elements that represent the response in mixing ratios to the emission fluxes (see Eq. 5). Since we do not have an adjoint model, these sensitivity functions are derived by perturbing each element of the emission flux field $\boldsymbol{\phi}$ over the target region by small increment and applying the forward model (WRF-GHG) to obtain the resulting perturbed concentration field ($\mathbf{C} + \Delta\mathbf{C}$) over the target region. Hence, $\mathbf{K}$ is calculated as follows:

$$\mathbf{K} = \frac{\mathbf{C} + \Delta\mathbf{C} - \mathbf{C}}{\sum_{TR} \boldsymbol{\Phi}_{perturbed} - \sum_{TR} \boldsymbol{\Phi}} \tag{9}$$

The posterior estimate of the scaling factor, $\hat{\boldsymbol{\lambda}}$, is derived by minimizing the cost function, $J(\boldsymbol{\lambda})$, as given in Eq.7.

### 3.3. Error covariance matrices

Bayesian inversion utilizes error covariance matrices to account for the measurement error and the prior flux error variances and co-variances. The measurement error covariance matrix, $\mathbf{S}_e$, is constructed by specifying the $XCO_2$ random errors (single measurement precision) derived using the error parameterization scheme described in Sect. 3.1. Note that the $XCO_2$ random error is primarily determined by the instrument signal-to-noise performance (but also to some extent by the retrieval algorithm, see Buchwitz et al. (2013a)) and is typically about 1.2 ppm (for the assumed threshold requirement signal-to-noise ratio performance assumption used by Buchwitz et al., 2013a) except for some especially unfavorable conditions such as low albedo and high SZA scenarios. Transport model uncertainty





is neglected here since the objective of current study is to quantify the uncertainty in the retrieved fluxes due to CarbonSat's retrieval errors only.

The prior flux uncertainty, $\mathbf{S}_{prior}$, is set uniformly to 40 % of the total emission over the target region to ensure that the difference between the "true" and prior fluxes is appropriately considered. We consider the fact that the
5 increased variability of emissions at the high resolution (as it is used in this study) leads to increased uncertainty due to the lack of information about the emission processes at the required spatial and temporal resolutions. The magnitude of $\mathbf{S}_{prior}$ is specified here based on the approximate difference between the IER and the EDGAR inventories over the target region. Any error correlations are neglected; hence $\mathbf{S}_{prior}$ is set to be a diagonal matrix.

## 4. Results: Estimation anthropogenic $XCO_2$ enhancement and retrieved flux uncertainty over Berlin

10 In this study, we use anthropogenic $XCO_2$ enhancement, which is defined as the enhancement in $XCO_2$ resulting from local anthropogenic emissions relative to the background concentration. The tagged tracer option in WRF-GHG stores $XCO_2$ enhancement resulting from EDGAR emissions separately, and we use this field to represent anthropogenic $XCO_2$ enhancement. The uncertainty in the retrieved emission attributed by CarbonSat's retrieval error is a function of the anthropogenic $XCO_2$ enhancement over Berlin, the number of potential observations in and
15 around Berlin, and the retrieval uncertainty (random and systematic components). In this manner we take into account the influence of these parameters to achieve a robust estimation of the retrieved surface emission uncertainty or error.

### 4.1. Local anthropogenic $XCO_2$ enhancement

The $XCO_2$ enhancements resulting from anthropogenic emissions over Berlin are estimated in order to assess
20 whether these emission enhancements are detectable by an instrument having the performance of CarbonSat i.e. to assess whether the resulting plumes are statistically significant and robust, thereby enabling the changes or trends in anthropogenic emission over the cities.

Figure 5 shows the "true" anthropogenic $XCO_2$ enhancement on a reference day (24[th] June 2008), the anthropogenic $XCO_2$ enhancement based on the IER inventory, and the difference in $XCO_2$ enhancement due to the difference in
25 emission inventories. From Fig. 3a and Fig. 5a, it can be concluded that, given the availability of a satellite instrument which is able to precisely detect the associated $XCO_2$ mixing ratio enhancements ranging from 0.80 to 1.35 ppm at a high spatial resolution and adequate spatial coverage, anthropogenic emissions from a city the size of Berlin and other localized emission sources can be estimated from space with sufficient accuracy. It should be noted that the magnitude of detectable anthropogenic $XCO_2$ enhancements is likely to be underestimated in our study
30 because the "true" fields of $XCO_2$ variations are simulated at a 10 km spatial resolution instead of CarbonSat's resolution (~2 km × 2 km).

Noteworthy is that the spatial and temporal difference in EDGAR and IER emission inventories gives rise to a notable $XCO_2$ mixing ratio difference between 0.4 and 1.0 ppm. For Berlin, this is about 40% of the total "true" $XCO_2$ enhancement. It should be noted that surface concentrations show larger relative differences than the column
35 dry mole fraction for $CO_2$, $XCO_2$, because of their higher sensitivity to the change in surface fluxes. Hence this result indicates the importance of characterizing emission uncertainties, even for the region where fossil emissions are often considered to be "well-quantified" in comparison to the biospheric carbon balance. Neglecting this uncertainty term would lead to significant biases in the net carbon exchange estimations, particularly when assimilating concentration measurements closer to emission sources such as cities.

40 ### 4.2. Uncertainty of the retrieved Berlin emissions



In this section, we show the results obtained by inverting CarbonSat simulated observations over the target region, taking into account different sources of possible errors including CarbonSat measurement errors and modeling errors. Inversions are performed separately for each potentially useful CarbonSat overpass (see above) to derive the total emission flux and its error over the target region.

The systematic error (SE) of the retrieved emission fluxes, which are specific for each source of errors or combination of errors, are determined separately by defining six scenarios represented by S01 through S06 (Table 1). These scenarios are described in the following subsections, while additional scenarios S07 – S11 are presented and discussed separately in Sect. 4.3. Note that the distance from the center of the target region to one of its boundaries is roughly 50 km, which corresponds to a time of approximately 3 hours for air parcels travelling with a

velocity of 4.5 ms$^{-1}$. This means that the observed local $CO_2$ emission plume is not only determined by the emission at the time of the overpass, but also during a time interval of several hours before the time of the overpass. This is taken into account when modeling the $CO_2$ emission plume. For the inversion, it is assumed that the time dependence of the emissions in the time period of up to several hours (3 to 6 hours) before the overpass is at least reasonably well known except for the scenarios S07 to S11. As noted earlier, the "true" $XCO_2$ variations in this

study are based on 10 km spatial resolution instead of 2 km in CarbonSat simulated observations. For the inversion results, we assume negligible representation error arising from these spatial scale mismatches. Based on meteorological conditions, the representation error introduced by decreasing the horizontal resolution from 2 km to 10 km can be approximately 0.5 ppm on average for $CO_2$ concentrations at the surface (Tolk et al., 2008). However, it is expected that the representation error for $XCO_2$ between these horizontal scales will be much lower than that for

$CO_2$ concentration at surface (see Pillai et al., 2010).

Before analyzing SE for the different scenarios, we first present the random error (RE) of the retrieved emission. RE is caused by the measurement noise, i.e., by the random part of the measurement error; hence it is independent of the above-mentioned SE scenarios. In the optimal case, the instrument noise is determined by the shot noise of the detector arrays. In practice, there are additional sources of noise such as read out noise, digitization noise etc. Figure

7 shows the random errors of the retrieved emissions over the target region, obtained by inverting the entire one-year data set of simulated CarbonSat $XCO_2$ retrievals. As explained above, we have investigated two different swath widths, 500 km and 240 km. The results are shown only for the days where the number of CarbonSat simulated observations around the target region is sufficiently dense (covering the emission plume and its surroundings) to obtain a retrieved emission random error of less than 25 %, i.e., we use the a posteriori random error of the retrieved

emission as a quality criterion (as also done in Buchwitz et al. (2013b)). This number, labeled as "N" useful overpasses, is 25 for a swath width of 500 km and 17 for a swath width of 240 km. As can be seen in Fig. 7, decreasing the swath width not only reduces the number of useful overpasses, but also increases the RE of the retrieved fluxes for some overpasses. The RE of the retrieved emission (from a single overpass) is usually found to be less than 20% (approximately 10 $MtCO_2$ yr$^{-1}$) of the emission fluxes for both swath widths.

**4.2.1. Impact of CarbonSat measurement errors (scenario S01)**

Here, we focus on scenario S01, and estimate the uncertainty in the retrieved emission fluxes caused exclusively by CarbonSat measurement errors. For this, we assume that the $XCO_2$ variability in the target region is dominated by the anthropogenic $CO_2$ emission and that there is negligible $XCO_2$ variability due to biogenic fluxes over the target region, or that this biogenic component can be modeled well, and thus can be subtracted from the observations

without introducing any modeling-related errors.

The systematic measurement error of the CarbonSat simulated observations over the target region for a typical day (24$^{th}$ June 2008) for S01 is shown in Fig. 8a. This is estimated using the error parameterization scheme of Buchwitz et al. (2013a), as shortly described in Sec. 3.1. The mean systematic measurement error over the target region is about 0.25 ppm for this day. For the scenario S01, the "observed" anthropogenic $XCO_2$ by CarbonSat is thus the

sum of this measurement error (Fig. 8a) and the "true" anthropogenic $XCO_2$. Fig. 9a shows the observed





anthropogenic $XCO_2$ enhancement for S01 over the target region during the overpass on 24[th] June 2008. For the comparison, the corresponding "true" anthropogenic $XCO_2$ enhancement, i.e. without any source of errors, is shown in Fig. 9g. The "true" emission plume, originating almost from the centre of the target region, can be clearly seen with a maximum value of about 0.90 ppm. As can be seen, the observed CarbonSat $XCO_2$ pattern (Fig. 9a) differs

from the "true" $XCO_2$ pattern (Fig. 9g) by the measurement errors (Fig. 8a); hence the retrieved emission via inversion typically differs from the "true" emission that results in a systematic error (SE) of the retrieved emission. The extent of this systematic error depends on how well the systematic measurement error correlates with the "true" $XCO_2$ pattern.

Figure 10 shows the systematic errors of the retrieved emissions for CarbonSat overpasses over the target region
obtained by inverting the entire one-year data set of simulated CarbonSat $XCO_2$ retrievals for the scenario S01. Shown are the results for swath widths of 500 km and 240 km for all "N" useful overpasses (days). Overall, the absolute magnitude of the systematic errors of the retrieved emissions for both swath widths for the scenario S01 is found to be less than 10% for most of the overpasses (about 75% of the "N" useful overpasses for the year 2008), which corresponds to about 5.3 $MtCO_2$ $yr^{-1}$. For the 500 km swath width in S01, the mean and standard deviation of
the SE for all "N" useful overpasses is -2.4 $MtCO_2$ $yr^{-1}$  (-4.5%) and 3.2 $MtCO_2$ $yr^{-1}$  (6.2%), respectively (see also Table 1). In general, we find that the two different swath widths have a negligible impact on the daily SE of the retrieved emissions.

### 4.2.2. Impact of CarbonSat measurement errors with worst-case aerosol related biases (scenarios S02 and S04)

Note that in the previous section we have used the CarbonSat systematic $XCO_2$ retrieval errors as provided by the error parameterization scheme described in Buchwitz et al. (2013a). However, as explained in Buchwitz et al. (2013b), this scheme may underestimate aerosol related biases if the spatially (not aggregated) high-resolution CarbonSat simulated observations are used for applications like the one used here. The reason is that aerosol-related retrieval biases have been computed using quite smooth model aerosol input data sets, which might not be sufficient
to represent the aerosol plume over Berlin.

To consider this, an additional error term has been defined which is referred to as "high resolution aerosol error" in this manuscript. In this sub-section we present results for scenario S02, where the measurement error used for S01 described in the previous section has been replaced by the high-resolution aerosol error contribution to the systematic measurement error. We also present results for scenario S04, where the measurement error is the sum of
the S01 and S02 errors.

The method of computing the "high resolution aerosol error" is described in detail in Buchwitz et al. (2013b). Here we describe it briefly as follows. A local AOD enhancement has been computed by scaling the observed anthropogenic $XCO_2$ spatial pattern, i.e., the AOD enhancement is assumed to be perfectly correlated with the $CO_2$ emission plume of interest (see Fig. 8b and Fig. 9b). Furthermore, a quite high scaling factor has been used (the
AOD change, $\Delta AOD$ at 550 nm is 0.2 per 4 ppm of local anthropogenic $\Delta XCO_2$). Overall, these are worst-case assumption that are supposed to result in upper limits of systematic $XCO_2$ errors due to aerosols and resulting errors of the retrieved emissions. For a more detailed discussion see Buchwitz et al. (2013b).

The resulting SEs of the retrieved emissions for scenario S02 are found to be negative, indicating systematic underestimation of retrieved emissions (see Fig. 11). As can be seen, the absolute magnitudes of errors are slightly
higher than those for S01. The mean and standard deviation of SE for S02, considering all "N" useful overpasses and the 500 km swath width, are -4.0 $MtCO_2$ $yr^{-1}$ (-7.3%) and 2.3 $MtCO_2$ $yr^{-1}$  (3.1%) respectively.

Another scenario, S04, investigates the impact of both high-resolution aerosol related errors (used for S02) and the "default" CarbonSat measurement errors (used for S01) on retrieving anthropogenic emissions. Inversions are





performed by utilizing these two sources of error, i.e. the $XCO_2$ systematic error for S04 is the sum of $XCO_2$ systematic error specified in S01 and S02 (see Fig(s). 8d and 9d)). As expected, the SE of the retrieved emission for S04 is found to be higher than those of S01 and S02, and their values are close to the linear sum of systematic emission errors for S01 and S02 (see Table 1). As already explained, the definition of S04 likely represents the possible worst-case measurement scenario in particular with respect to aerosol related errors.

### 4.2.3. Impact of biospheric modeling error (S03, S05, S06)

In this section, we explore the impact of modeling error on retrieving Berlin city emissions. In the last two sections, it is assumed that the spatial variability introduced by the biogenic component of $XCO_2$ in the target region is well known or sufficiently small that it can be ignored. However, in reality there are notable perturbations caused by the spatial variability of biogenic $XCO_2$ in the target region that cannot be ignored. As an example, Fig. 8c illustrates the biogenic $XCO_2$ variability in the target region during a CarbonSat overpass. Most critical in terms of this uncertainty is how well the biogenic $XCO_2$ pattern is correlated with the anthropogenic $XCO_2$ pattern. In this case, the uncertainty in the retrieved emissions depends on how accurately the biogenic fluxes can be modeled, as well as the associated transport model uncertainty in simulating the biogenic $XCO_2$ pattern. Note that we assume negligible transport uncertainty for the anthropogenic $XCO_2$ pattern in order to distinguish the retrieved emission errors due only to the biogenic $XCO_2$ pattern. In order to account for this modeling-related error, we consider scenario S03. In S03, we assume an extreme case where biogenic $XCO_2$ cannot be modeled at all; hence biogenic $XCO_2$ is treated as the "perturbation" seen in the measurement vector ($y$) of the inversion system (see Fig(s). 8c and 9c). However, it should be noted that in reality biospheric modeling uncertainty is not expected to be as high as this assumption. As can be seen in Fig. 2, a simple biosphere model such as VPRM used in this study could capture 50 to 65% of the biospheric flux variability in most of the cases (squared correlation coefficient (VPRM vs. observations), $R^2 \sim 0.50$-$0.65$).

The systematic errors of the retrieved emissions for S03 are found to be significantly higher compared to the errors for the above-mentioned scenarios than those for S01 and S02 (see Fig. 12). Noteworthy is that this uncertainty is not related to CarbonSat measurement errors, but arises due to the inability of the model to simulate the biospheric contribution. Hence this uncertainty should be treated as model-related error. Due to the extreme assumption of modeling error in S03, the uncertainty values reported in this section have to be considered as the extreme upper limits of the possible total uncertainties in the retrieved fluxes due to biogenic modeling error. Despite this, the SE of the retrieved emission for S03 is within the range of 20 to 25% (10 to 15 $MtCO_2$ $yr^{-1}$) for most of the scenes although we assumed the largest uncertainty in modeling biogenic $XCO_2$. The reason for this is that the spatial biospheric $XCO_2$ pattern in the target region that "disturbs" the inverse system typically differs from the anthropogenic $XCO_2$ pattern in many of the good CarbonSat overpasses, enabling these two sources/sinks (anthropogenic and biogenic) to be disentangled reasonably well.

Additionally, we define other scenarios, S05 and S06, to investigate the impact of the biogenic modeling errors in combination with other error sources, such as CarbonSat measurement errors and high-resolution aerosol related errors. Systematic error estimations for these scenarios are summarized in Table 1 and these results suggest that a dominant part of the retrieved emission error is caused by the unknown biogenic variability.

### 4.3. Inversion Experiment using different prior emission fluxes (S07-S11)

The inversion results presented so far have not taken into account the impact of imperfect knowledge of the spatial pattern of emission fluxes and the different time dependences of the emissions; hence the inverse optimization adjusts only the amplitude of the emission plume corresponding to the anthropogenic $CO_2$ emission in the target region. Although the error arising from these unknown spatial emission structures is not directly related to CarbonSat measurement errors, we attempt to perform an experiment using two different flux inventories, with one of the flux inventories representing the prior fluxes and the other as the "true" fluxes. The experiment is designed





with an inversion set-up, which is essentially the same as that described in Sect. 3.3, but with the following exception: Here the prior emission fluxes are prescribed from the IER emission inventory (Fig. 3b); hence the modeled anthropogenic $XCO_2$ is based on IER emission fluxes (see Fig. 5b and Sec. 2.1.1). Similar to the sections above, the EDGAR emission inventory is considered to be the "true" fluxes and the measurement vector ($y$) which

corresponds to CarbonSat simulated observations is based on the EDGAR emission inventory as described in sec. 3.1. The retrieved posterior fluxes of this inversion optimization are compared with "true" fluxes to estimate the retrieved posterior flux errors and to assess how well inversion studies can benefit from CarbonSat measurements in the case of discrepancy between "true" and prior fluxes in terms of spatial patterns of distribution.

Similar to the above section, systematic errors of the retrieved fluxes are estimated specifically for each source of

errors or combination of errors by defining scenarios S07 through S11 (see Table 1). It should be noted that the IER and EDGAR fluxes are not entirely different in terms of temporal variations, though the magnitude of the emissions in the target region is notably different. However, there exists a dissimilarity of approximately 70% of the spatial patterns between these two inventories (based on the correlation of spatial variability between two inventories, $R^2 \sim$ 0.30) in the target region.

For most of the overpasses, the random errors of the retrieved emission fluxes over the target region (single overpass) are found to be less than 20% (approximately 10 MtCO$_2$ yr$^{-1}$) of the emission fluxes for both swath widths (not shown). These values are comparable to those shown in Fig. 7, indicating the potential of CarbonSat simulated observations to retrieve surface fluxes, even when uncertainties in the spatial pattern of the prior emission fluxes are present. Figure 13 shows the SE of the retrieved emissions estimated for the scenario S08, where CarbonSat

measurement errors are considered in addition to the uncertainty in the spatial pattern of the prior fluxes. For both swath widths, the estimated SE for S08 is found to be less than 10 MtCO$_2$ yr$^{-1}$ in many instances (for about 55 to 75% of useful overpasses). Systematic errors for other scenarios are summarized in Table 1. Depending on the error sources, the inversion experiment shows that the mean and standard deviation of SE, considering all "N" useful overpasses and the 500 km swath width, ranges from -0.12 to -9.0 MtCO$_2$ yr$^{-1}$ and 14.6 to 19.2 MtCO$_2$ yr$^{-1}$

respectively. Furthermore, the systematic errors of the retrieved emission fluxes for both swath widths are found to be lower than the systematic error of the prior fluxes (estimated based on "true" fluxes) except for a very few cases, providing confidence in the inverse results although only a simple inverse optimization methodology is used.

## 5. Discussion and clean pixel method

In this section, we discuss the merits of instrument like CarbonSat for retrieving emission fluxes and its potential in

disentangling anthropogenic and biogenic $CO_2$ fluxes over cities like Berlin. Caveats related to the simple inversion approach used here are discussed.

For the study of $CO_2$ emissions, it is necessary to assess whether local anthropogenic $XCO_2$ enhancements are large enough to be detected by using the retrieved $XCO_2$ data products from the satellite-borne instrument, taking into account the measurement noise. Figure 14 presents an overview of "true" anthropogenic $XCO_2$ emission

enhancements together with the associated CarbonSat retrieval uncertainty over the target region around Berlin for a one year time period. The analysis shows that anthropogenic $XCO_2$ enhancements around Berlin are well above the retrieval biases for most of the overpasses and the number of potential observations, after filtering out the contaminated pixels, is large enough to minimize the random error component. Given the availability of such a dense sampling coverage with similar retrieval biases, one can be confident in utilizing CarbonSat's observations for

retrieving city emission trends or absolute emission fluxes via appropriate inverse modeling.

In a real scenario the question arises whether it is possible to clearly separate local anthropogenic $XCO_2$ enhancements from CarbonSat's total column measurements, which are in addition influenced by biospheric sources or sinks. Moreover, in order to isolate the $XCO_2$ enhancement caused by local sources (such as city emissions), it is necessary to specify the "background" signal, representing the $CO_2$ column without any influence of local fluxes.



These additional biospheric and background influences can be ignored if the target city is well isolated from other strong urban sources and/or active biospheric regions as well as has negligible local biospheric activity. However, only a few cities or urban areas meet the above criteria, and a typical European city, in general, has considerable local or nearby biogenic influences. Under these conditions it is necessary to disentangle biogenic, anthropogenic

and background contributions from CarbonSat's observations. To assess the relative contribution of biogenic and anthropogenic sources, one can utilize additional tracers (e.g. CO, $NO_X$) and/or isotopic ratios (e.g. $\Delta^{14}C$) as demonstrated by Newman et al. (2013). In the time frame of a potential CarbonSat mission, Sentinel-5 will be providing data on CO and tropospheric $NO_2$ (Ingmann et al., 2012), which when combined with CarbonSat data allows for the attribution of air masses originating from fossil fuel combustion. Depending on the extent of the

variability and the possible uncertainties, we can also rely on the biospheric and global model simulations to differentiate different source-sink contributions.

In this study, we describe an approach to estimate the anthropogenic Berlin $XCO_2$ enhancements from measurements made by and data products retrieved from the proposed instrument CarbonSat. In this manner, we assess its capability to track the anthropogenic enhancements in our target region and thereby retrieve or infer

emissions. As our target region around Berlin is mostly isolated from other urban sources, we use a simple "clean-pixel" method, similar to that used by Kort et al. (2013) to differentiate local anthropogenic $XCO_2$ from other background and biospheric surface fluxes. We have chosen boundary pixels of our target region in the upwind direction as "clean pixels", assuming that the observations from these pixels typically represent background $XCO_2$ values without any local influence. The WRF simulated wind direction in the lower atmosphere yields the upwind

direction of the target region. Berlin $XCO_2$ enhancements are estimated by differentiating these plumes in the simulated CarbonSat observations over the target region (see Fig. 14). As is shown, the temporal patterns of the estimated anthropogenic enhancement are in general consistent with those of the true anthropogenic enhancement. This approach is thus able to isolate anthropogenic $XCO_2$ enhancements with a mean bias (mean of the difference between estimated and "true" enhancements) of 0.12 ppm, a standard deviation of the difference $\sigma_d = 0.17$ ppm, and

a squared correlation coefficient $R^2 = 0.92$.

One source of bias on the $XCO_2$ enhancement estimated by the clean-pixel method is when the dominant biogenic perturbations in the target region have different patterns than those of the chosen "clean" boundary region. Another possibility is the failure of the clean-pixel method to represent background $XCO_2$ concentration based on clean boundary pixels. The tagged tracer option in WRF-GHG allows us to investigate this further by utilizing the

modeled biogenic and background $CO_2$ concentrations generated by WRF-GHG. We found that most of the deviations in the estimated $XCO_2$ enhancement are caused by the background "noise", indicating that the $XCO_2$ from the "clean" boundary pixels do not always represent the background values in our case. Note that the above-mentioned bias is not related to any CarbonSat measurement errors, but due to the simplicity of the approach adopted to estimate anthropogenic $XCO_2$ enhancements. We found negligible influence of biospheric fluxes in the

target region, which can bias the $XCO_2$ enhancement estimated by the clean-pixel method (not shown).

By assuming that the biospheric patterns are accurately modeled and that these biogenic signals can be subtracted from the measurement vector to isolate the anthropogenic contribution of $XCO_2$, our simple inversion system is constructed such that it takes into account the impact of CarbonSat sampling errors on the retrieved city emissions over Berlin. The applicability of our results to a scenario where these assumptions are not valid needs to be

examined, but the current set-up is not well suited for this purpose since we have not taken into account additional state vectors for biospheric contributions. On the other hand, the current setup allows us to investigate the extremely pessimistic scenario where we assume that we cannot model the biospheric contribution at all (see Sec. 4.2.3).

Although we utilize high-resolution forward simulations, at present our inversion system uses only one scaling factor for the entire target region for each useful overpass. This means that the current set-up cannot provide

posterior estimates for each pixel or emission sector within the target region. In other words, the flexibility to capture the true spatial variation of fluxes is more limited in our simple inversion system than in pixel- or parameter-





wise inversions. Using this simple inversion system may thus overestimate the retrieved flux uncertainty. While interpreting our results, one should keep in mind that we do not specify other important sources of errors in the inversion system such as transport error. As previously noted, the main focus of this study is to estimate the retrieved flux uncertainties that are caused only by CarbonSat's measurement errors. However, these transport

related errors, which provide proper weight to the observations depending on the capability of the transport model, need to be taken into account when estimating the total flux uncertainty via inverse modeling.

## 6. Conclusion

In the present study, we examine the potential of a satellite mission like CarbonSat for improving the current knowledge on the surface-atmosphere exchange of atmospheric $CO_2$. A significant contribution by the CarbonSat

greenhouse gas (GHG) observations will be the ability to retrieve the emissions of localized (moderate to strong $CO_2$ and $CH_4$) emission sources such as cities, power plants, methane seeps, etc., as a result of its unique sampling capability at high spatial resolution (approximately 2 km × 2 km) with a good spatial coverage using a much wide swath. To demonstrate this, we have simulated emissions from a medium-size city (in terms economic contribution and trade) and assessed the capability to retrieve anthropogenic emission fluxes for the city and its surrounding

region (Berlin-centered target region investigated here: ~100 km × 100 km) from CarbonSat simulated observations.

The study utilizes a Bayesian inversion approach based on the WRF-GHG modeling system at a high spatial resolution to optimize anthropogenic $CO_2$ emissions for the target region using CarbonSat simulated observations for a time period of one year. The inverse system is designed in such a way that one can quantify the random and systematic errors of the retrieved anthropogenic emission fluxes for a given set of $XCO_2$ measurement and modeling

errors. The CarbonSat measurement errors are estimated using the error parameterization scheme of Buchwitz et al. (2013a), which takes into account different sources of uncertainties including scattering related errors. Based on the EDGAR emission inventory, the local anthropogenic $XCO_2$ enhancement over Berlin is found to be approximately 0.80 to 1.35 ppm. The latter is similar to the detectable limit of single CarbonSat ground pixels. However typically there will be several hundred observations available per overpass, sampling the emission plume and its surrounding.

The impact of CarbonSat measurement errors on the retrieved emissions is assessed for two swath widths (240 km and 500 km). By performing a Bayesian inversion based on one year of CarbonSat simulated observations, we show that the random error of the retrieved Berlin $CO_2$ emissions is typically less than 15 to 20% of the total city emissions. In other words, the CarbonSat measurements can be utilized in atmospheric top-down approaches to quantify emissions of medium sized cities such as Berlin with a precision better than 8 to 10 $MtCO_2$ $yr^{-1}$.

In order to quantify the systematic error (SE) of the retrieved fluxes, we use different scenarios in terms of various sources of systematic error in the inversion system. For scenario S01, we use CarbonSat's "default" $XCO_2$ systematic errors (retrieval biases) from Buchwitz et al. (2013a), and assume no biogenic $XCO_2$ modeling error. For S01, we find that SE is in the range of 3 to 6 $MtCO_2$ $yr^{-1}$ for most of the cases (40 to 80% of the "good" overpasses as identified by the quality filtering procedure), indicating a high potential of utilizing CarbonSat's measurements to

retrieve city emissions. Based on the analysis using a one-year period of CarbonSat simulated observations, we show that narrowing the swath width (from 500 km to 240 km) decreases the total number of useful overpasses, as expected, but we do not find any significant difference between the single overpass SEs estimated for the two swath widths investigated here.

As explained in Buchwitz et al. (2013b), the default $XCO_2$ systematic errors only reflect aerosol related biases at

quite low spatial resolution. On the spatial scale of the city of Berlin, aerosol-related biases may be larger. To consider this, we use the "worst-case" measurement scenario as used by Buchwitz et al. (2013b), in which we assume that the aerosol-related biases may be perfectly correlated with the signal of interest, which is the city $CO_2$ emission plume in combination with a high amount of aerosols in the plume. For this, we define a scenario S04 and refer to this as "high resolution aerosol error" in this manuscript. The estimated emission uncertainty for this





scenario (S04) is found to be higher than that of S01, with mean and standard deviation of approximately -6.4 and 4.2 $MtCO_2$ $yr^{-1}$ respectively.

The above-mentioned results, however, are mostly dominated by the assumption that there is a negligible influence of biospheric fluxes that perturb the emission plume over the target region, or that these biospheric contributions can be modeled very well. By further investigating the extreme case in which the biospheric contribution is assumed to be totally unknown and treated as perturbation in the inversion system (scenario S03), we find that the single overpass SE of the retrieved emission is significantly increased to $8.5 \pm 10.8$ $MtCO_2$ $yr^{-1}$ (mean $\pm$ standard deviation). Nevertheless, the magnitude of the uncertainty is not overwhelmingly large over the target region, despite the worst-case assumption used here. It should be kept in mind that the above-mentioned uncertainty is not directly related to the performance of CarbonSat measurements, but more towards the models' inability in simulating the biospheric contribution well. Hence, for the effective utilization of these measurements, the "noises" induced from other sources have to be taken into account, which requires careful design of the inverse optimization methodology using transport models at high resolution, enabling them to handle the information contained in those measurements. On comparing the results from different scenarios, we show that the systematic error of the retrieved fluxes depends largely on the accuracy of the CarbonSat simulated observations and more importantly on the modeling related errors.

Further investigation by designing a synthetic inversion experiment is motivated by the possible impact of spatial structural variability of the emission fluxes, which is not considered in the above-mentioned inversions. We acknowledge that our current inversion set-up is too simple to examine how suitable CarbonSat measurements are for this purpose, as we use only one scaling factor for the entire target region. Nevertheless we find promising results from this experiment in which the modeled and true $XCO_2$ concentrations are based on two distinct emission inventories (IER and EDGAR) differing in spatiotemporal patterns. By showing that the systemic error of the retrieved fluxes is lower than that of the prior fluxes (estimated based on true fluxes) in most of the cases, the results from the inversion experiment build confidence in our uncertainty estimations and ensure that the optimization is done correctly. The random error of the retrieved emissions for a single overpass is estimated to be less than 10 $MtCO_2$ $yr^{-1}$ for both swath widths. Hence it is expected that given the availability of the high-resolution CarbonSat simulated observations, it is likely to deduce the structural patterns of the emission fluxes. Based on the above analysis, however, no firm conclusion can be made on the magnitude of the retrieved flux uncertainty when prior fluxes significantly deviate from true fluxes in representing the structural variations of emissions. For this purpose, a more sophisticated inverse methodology involving additional extended state vectors and calculation of the response function of the elements of the state vector (adjoint calculation) is required. Since we use the same transport model to generate the (pseudo) observations and the influence functions, the inversion results shown here may be slightly optimistic. Although it is not within the scope of this study, the transport-related errors are expected to be non-negligible and should be properly addressed in the inverse modeling applications of satellite data.

Using the dense CarbonSat measurements in an inverse modeling framework at high resolution is expected to improve the inference of $CO_2$ fluxes by disentangling different sources of variations. But to what extent one can differentiate regional contributions from different sources should be investigated in further detail. A preliminary analysis over the target region using the "clean-pixel" method as followed by Kort et al. (2013) provides encouraging results in isolating temporal patterns of local anthropogenic $XCO_2$ enhancement.

Overall, the present study demonstrates that an instrument like CarbonSat has high potential to provide important information on city $CO_2$ emissions when exploiting the atmospheric $XCO_2$ observations using a high-resolution inverse modeling system. Utilizing these measurements together with in-situ, airborne and other satellite measurements is expected to provide more detailed and reliable information on natural and anthropogenic fluxes, facilitating the monitoring of future climate mitigation strategies.



***Acknowledgements.*** We thank all principle investigators involved in the eddy covariance measurements, and all scientists involved in the L4 eddy covariance dataset that has been accessed from http://www.europe-fluxdata.eu/. This study has received funding from ESA (projects LOGOFLUX-I and LOGOFLUX-II) and the State and the University of Bremen.



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





**Table 1:** Overview of different scenarios, SCE, which are used to investigate the systematic errors of the retrieved emissions. The absolute mean and standard deviations are estimated for two swath widths (SW-500: 500 km and SW-240: 240 km) for all "N" useful overpasses and are expressed in both $MtCO_2$ $yr^{-1}$ and in %. Err-L, Err-H, and Err-B indicate errors attributed to CarbonSat measurement, high-resolution aerosol related errors, and biogenic modeling errors respectively. Err-Emi indicates whether inversion experiment uses different prior emission fluxes (see Sec. 4.3)

| SCE | Err-L | Err-H | Err-B | Err-Emi | Prior flux | True flux | SE (SW-500) (mean ±std) % | SE (SW-500) (mean ±std) $MtCO_2$ $yr^{-1}$ | SE (SW-240) (mean ±std) % | SE (SW-240) (mean ±std) $MtCO_2$ $yr^{-1}$ |
|---|---|---|---|---|---|---|---|---|---|---|
| S01 | ✔ | | | | EDGAR | EDGAR | -4.5±6.2 | -2.4±3.2 | -5.6±5.4 | -3.0±2.9 |
| S02 | | ✔ | | | EDGAR | EDGAR | -7.3±3.1 | -4.0±2.3 | -6.8±3.0 | -3.6+1.9 |
| S03 | | | ✔ | | EDGAR | EDGAR | 19.3±23.9 | 8.5±10.8 | 21.2±24.5 | 9.3+10.8 |
| S04 | ✔ | ✔ | | | EDGAR | EDGAR | -11.9±7.1 | -6.4±4.2 | -12.4±6.8 | -6.5±4.1 |
| S05 | ✔ | ✔ | ✔ | | EDGAR | EDGAR | 7.4±24.9 | 2.1±11.8 | 8.8±26.1 | 2.8±12.1 |
| S06 | ✔ | | ✔ | | EDGAR | EDGAR | 14.8±24.5 | 6.1±11.2 | 15.6±25.4 | 6.3±11.3 |
| S07 | | | | ✔ | IER | EDGAR | -10.3±27.6 | -5.2±15.4 | -11.7±24.9 | -7.9±17.1 |
| S08 | ✔ | | | ✔ | IER | EDGAR | -11.9±27.6 | -6.0±15.7 | -15.1±22.7 | -9.6±16.2 |
| S09 | ✔ | ✔ | | ✔ | IER | EDGAR | -17.4±25.0 | -9.0±14.6 | -20.7±20.7 | -12.3±15.4 |
| S10 | ✔ | | ✔ | ✔ | IER | EDGAR | 1.2±33.8 | -0.12±19.2 | 2.4±31.6 | -1.1±20.4 |
| S11 | ✔ | ✔ | ✔ | ✔ | IER | EDGAR | -4.3±31.6 | -3.1±18.3 | -3.1±30.1 | -3.8±19.9 |





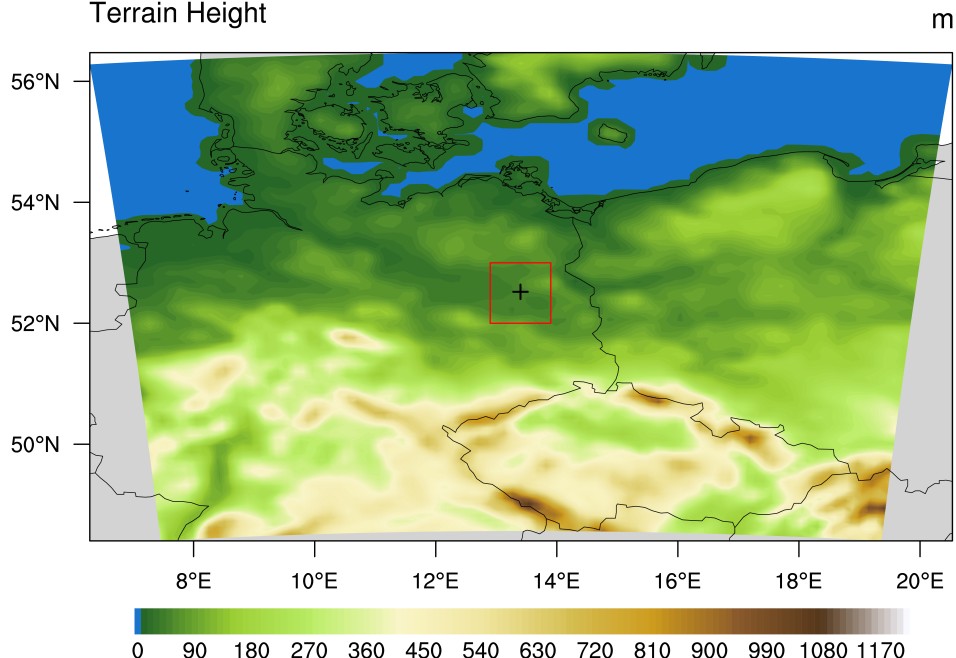

**Figure 1: The Berlin-centered WRF-GHG model domain in Lambert Conformal Conic projection used in the study. The red rectangle represents the target region (100 km x 100 km) described in the Sec. 3.2 and the + sign indicates the central location of Berlin city. The color bar indicates the terrain height in meters.**





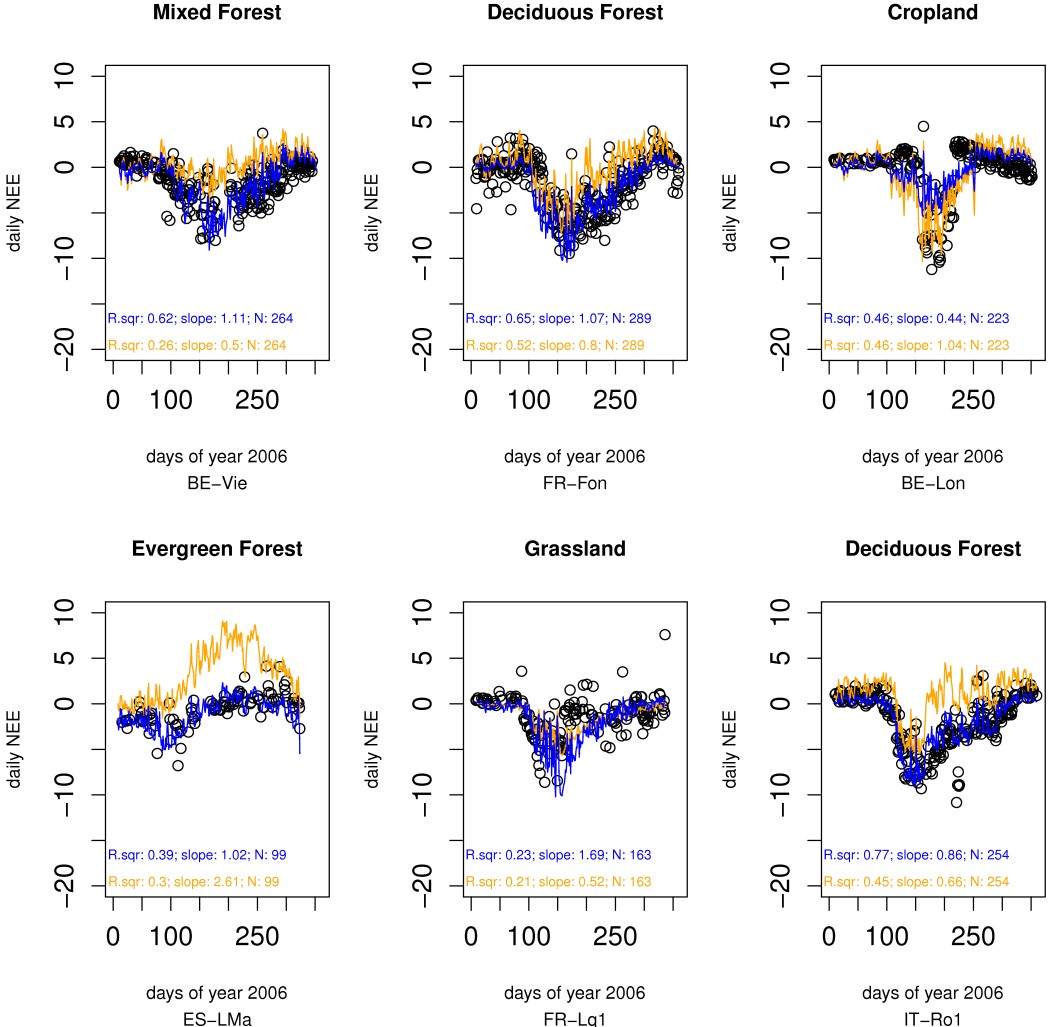

**Figure 2: Comparison between daily averaged carbon flux observations (Net Ecosystem Exchange, NEE) and the VPRM model simulations. The black circles represent observations, and the orange and blue curves denote VPRM simulations before and after optimization with flux data respectively.**



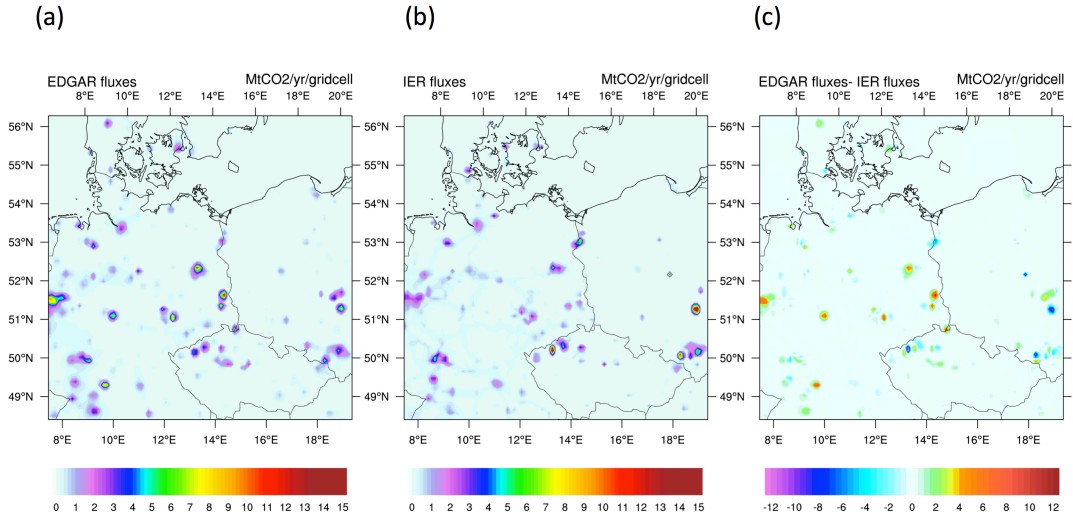

**Figure 3: Fossil fuel combustion emission fluxes over the model domain at a spatial resolution of 10 km × 10 km, averaged for the CarbonSat overpass periods during the year 2008: (a) EDGAR emissions, (b) IER emissions, and (c) the difference between EDGAR and IER emissions (EDGAR - IER). All units are in MtCO$_2$ per year per grid cell.**




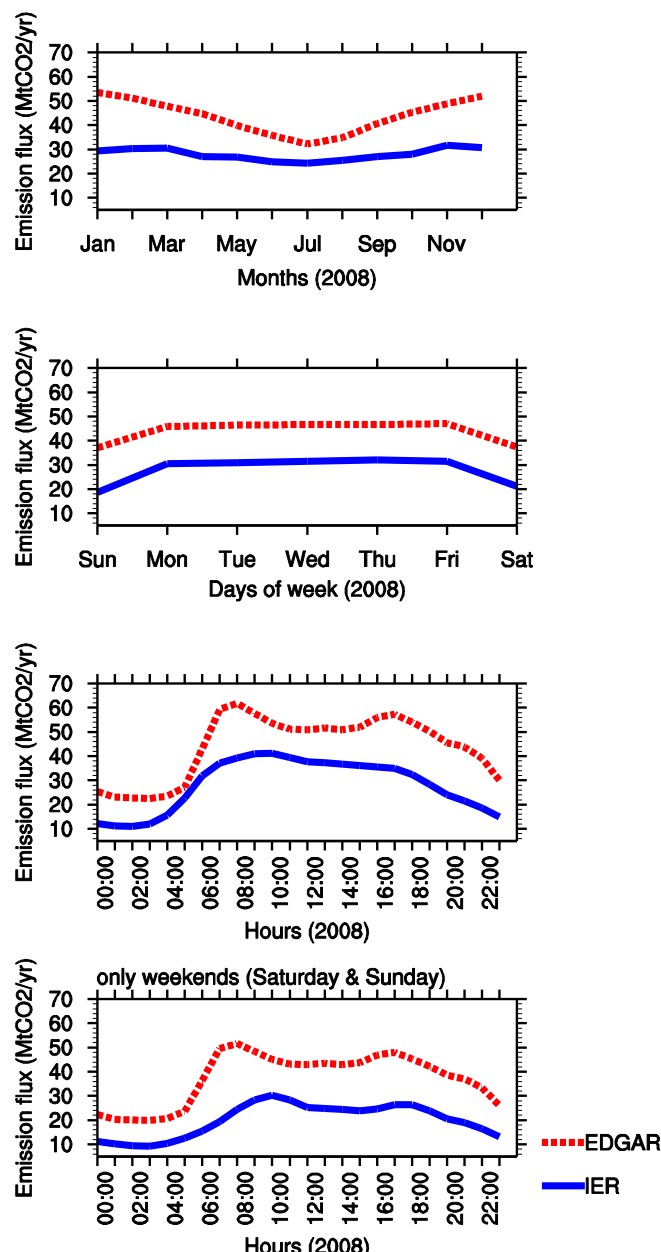

**Figure 4: Temporal variability of EDGAR and IER emission fluxes, aggregated over the target region around Berlin (~ 100 km × 100 km) averaged for different time scales for the year 2008: monthly (1ˢᵗ (top) panel), weekly (2ⁿᵈ panel) and hourly (3ʳᵈ and 4ᵗʰ panels). The 4ᵗʰ panel shows the values representing only weekends, while the 3ʳᵈ panel represents all days of the week. Hours are in UTC (local time CET = UTC+1).**



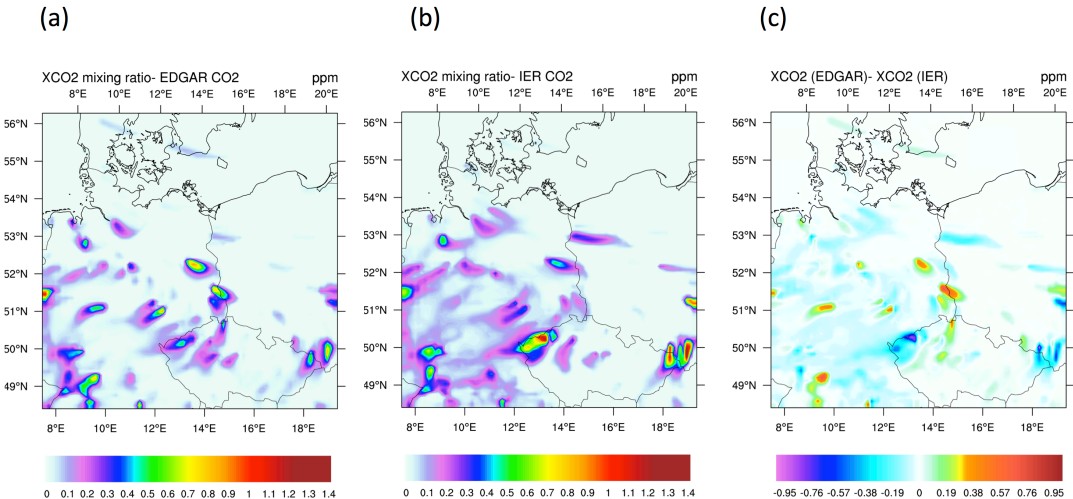

**Figure 5: Anthropogenic XCO$_2$ enhancement on 24$^{th}$ June 2008 at 10:00 UTC (Local time: 12:00 CEST) (a) "true" XCO$_2$ enhancement (using EDGAR emissions), and (b) XCO$_2$ enhancement when using IER emissions. Panel (c) shows the discrepancy in XCO$_2$ enhancement due to the difference between EDGAR and IER emission inventories. All units are in ppm.**





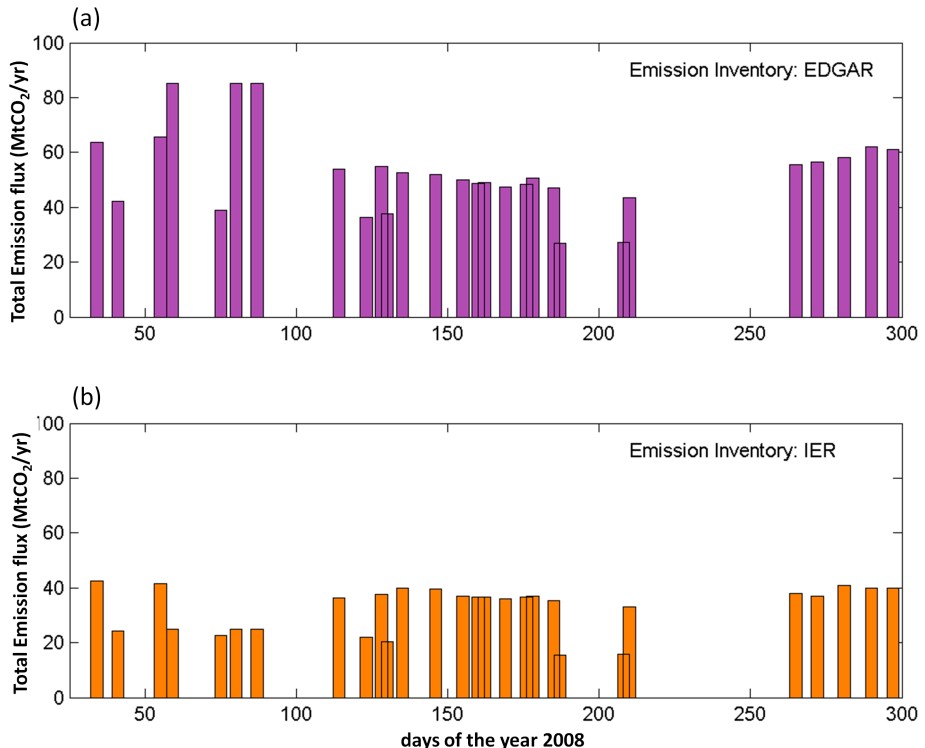

**Figure 6: Anthropogenic flux over the target region based on (a) EDGAR inventory, and (b) IER inventory for all of CarbonSat's "useful overpasses" corresponding to 500km swath width for the year 2008.**



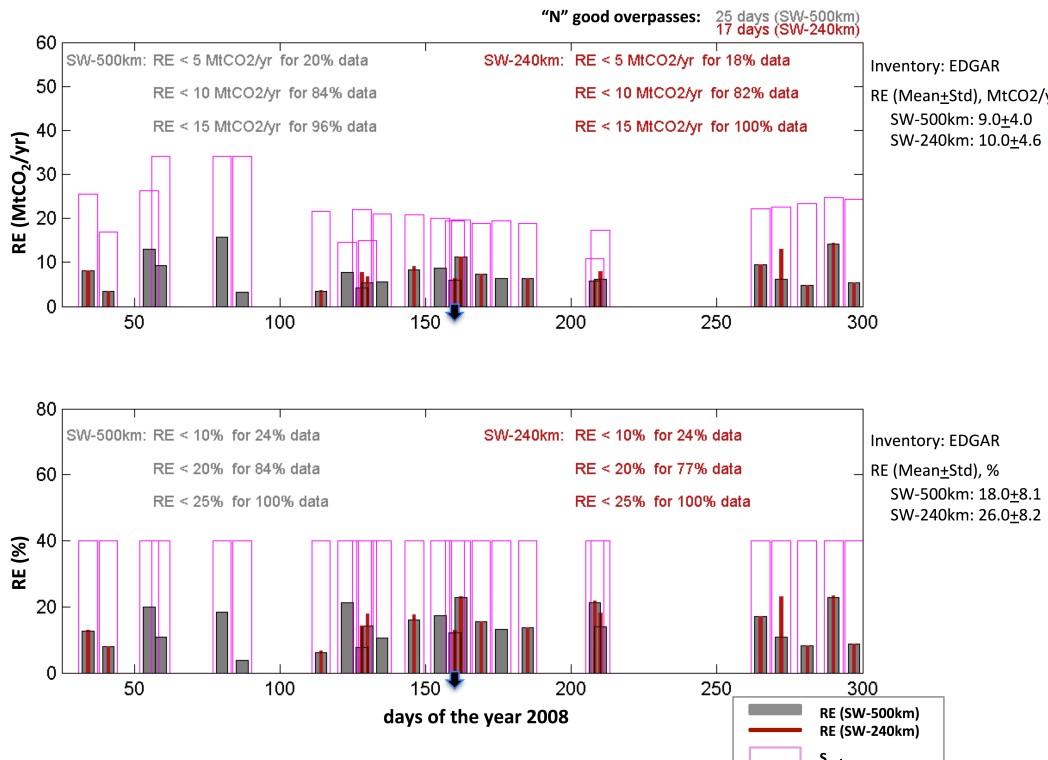

**Figure 7: Precision (random errors (RE)) of the retrieved emission fluxes obtained by the inverse optimization using one year of CarbonSat simulated observations. Results of two different swath widths (SW) – 500 km (grey) and 240 km (red) – are shown. $S_{prior}$ values are indicated with magenta bordered bars for visualizing the reduction in uncertainty. The top and bottom panels show RE in $MtCO_2 \ yr^{-1}$ and in % respectively. An overview of the statistical distribution of RE, separately for 500 km (grey) and 240 km (red) swath widths, is given inside the panel. The overall mean ± standard deviation is given outside the respective panels. The lower and upper limits of the X-axis (days of the year) is restricted accordingly as there are no good CarbonSat simulated observation during winter months. The arrow marker in the X-axis indicates a particular day ($24^{th}$ June 2008) shown in Figs. 5, 8, and 9.**





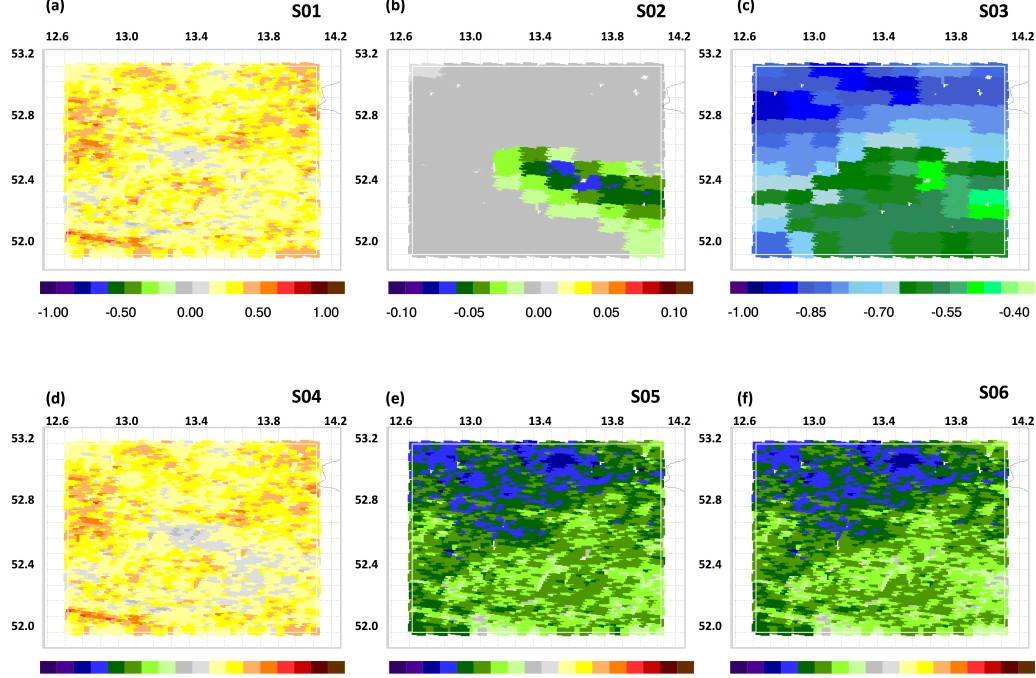

**Figure 8:** XCO$_2$ systematic error over the target region on 24$^{th}$ June 2008, assuming a CarbonSat swath width of 500 km. The six scenarios (S01 to S06) are shown with a label inside the respective panel. For S01, S02 and S04, these errors are estimated using the error parameterization scheme of Buchwitz et al. (2013a). The other scenarios additionally utilize biogenic XCO$_2$ variability in the target region (simulated by WRF-GHG) to derive XCO$_2$ systematic errors. Note that different color scales are used for S02 and S03. All units are given in ppm.



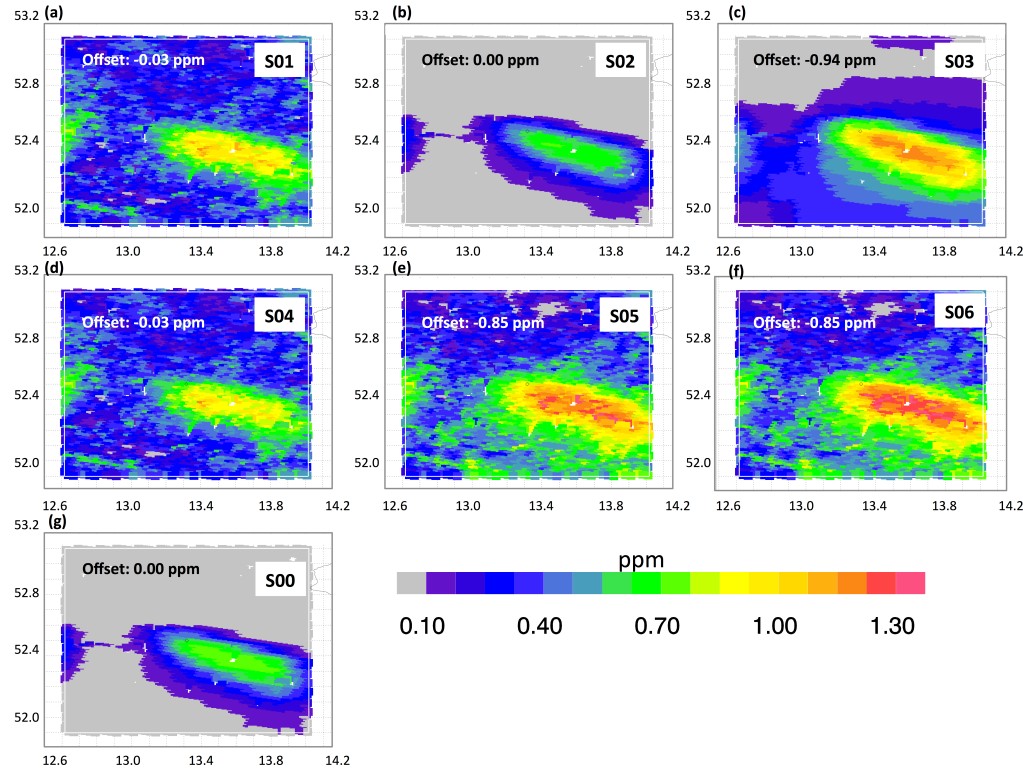

**Figure 9: Observed anthropogenic XCO$_2$ enhancement over the target region during a CarbonSat overpass on 24$^{th}$ June 2008 (swath width: 500 km). Different panels show anthropogenic XCO$_2$ enhancement, while considering XCO$_2$ systematic errors for different scenarios as shown in Fig.8. The "True" XCO$_2$ (fossil fuel (FF)) enhancement (i.e. without any uncertainties) is given in the bottom panel (g) for comparison. Note that an offset, labeled inside each panel, is subtracted from the anthropogenic XCO$_2$ enhancement to better visualize the details (for the figure only). All units are ppm.**



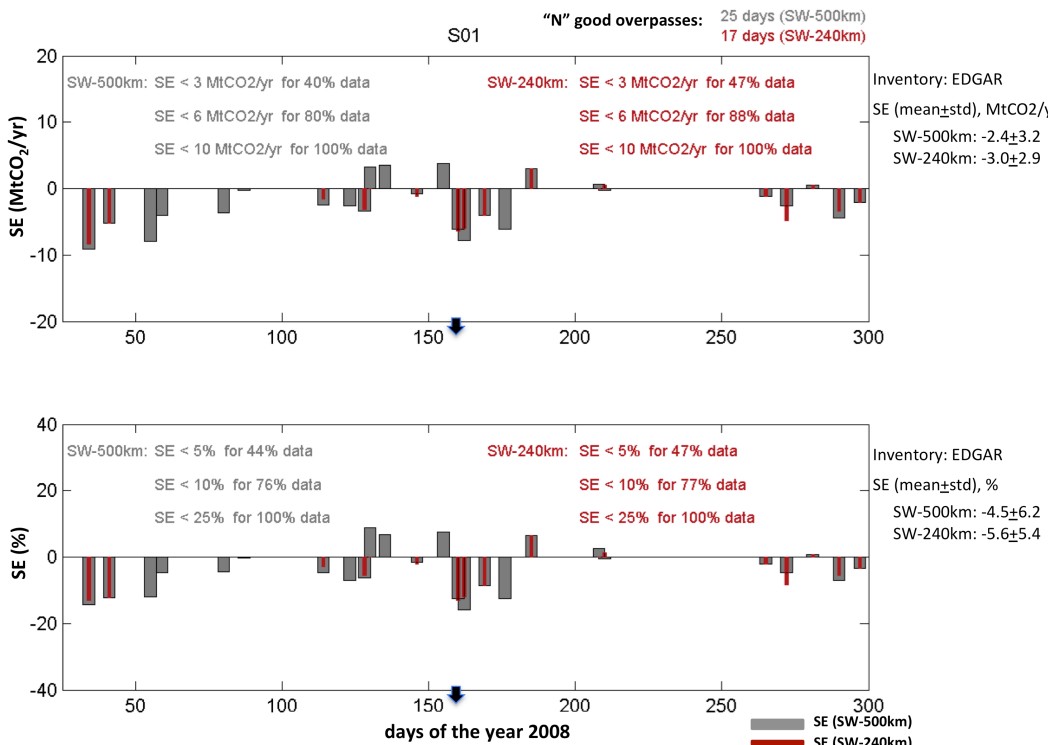

**Figure 10: Systematic errors (SE) of the retrieved emission fluxes for S01, obtained by the inverse optimization using one year of CarbonSat simulated observations. Results of two different swath widths (SW) – 500 km (grey) and 240 km (red) – are shown. The top and bottom panels show SE in MtCO$_2$ yr$^{-1}$ and in % respectively. An overview of the statistical distribution of SE, separately for 500 km (grey) and 240 km (red) swath widths, is given inside the panel. The overall mean +/- standard deviation is given outside the respective panels.**



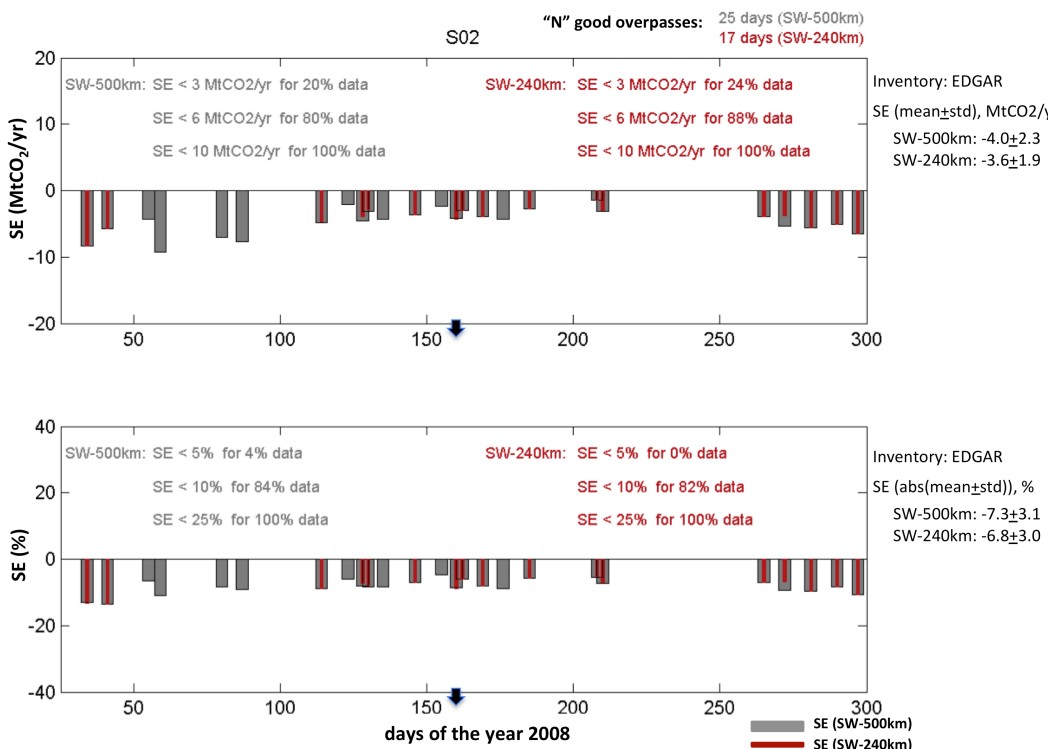

**Figure 11: Same as Fig.10, but for S02, quantifying the impact of the worst-case assumption used for aerosol-related biases.**




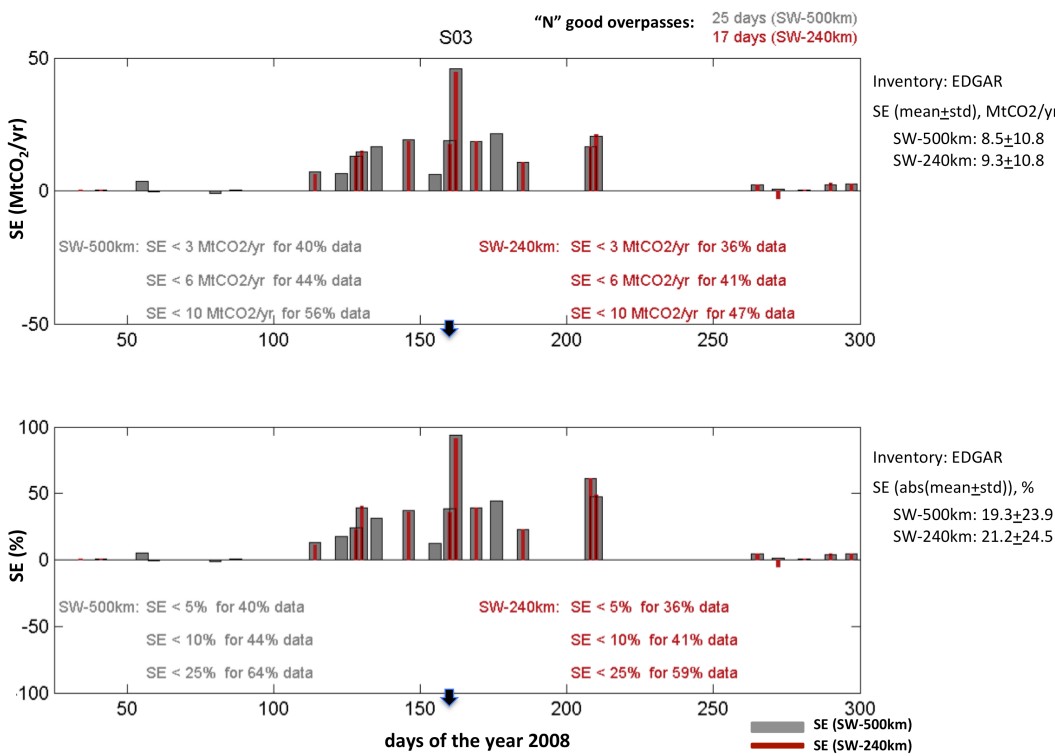

**Figure 12: Same as Fig.10, but for S03, quantifying the impact of the worst-case modeling related errors by assuming that biogenic XCO$_2$ variations cannot be modeled at all.**





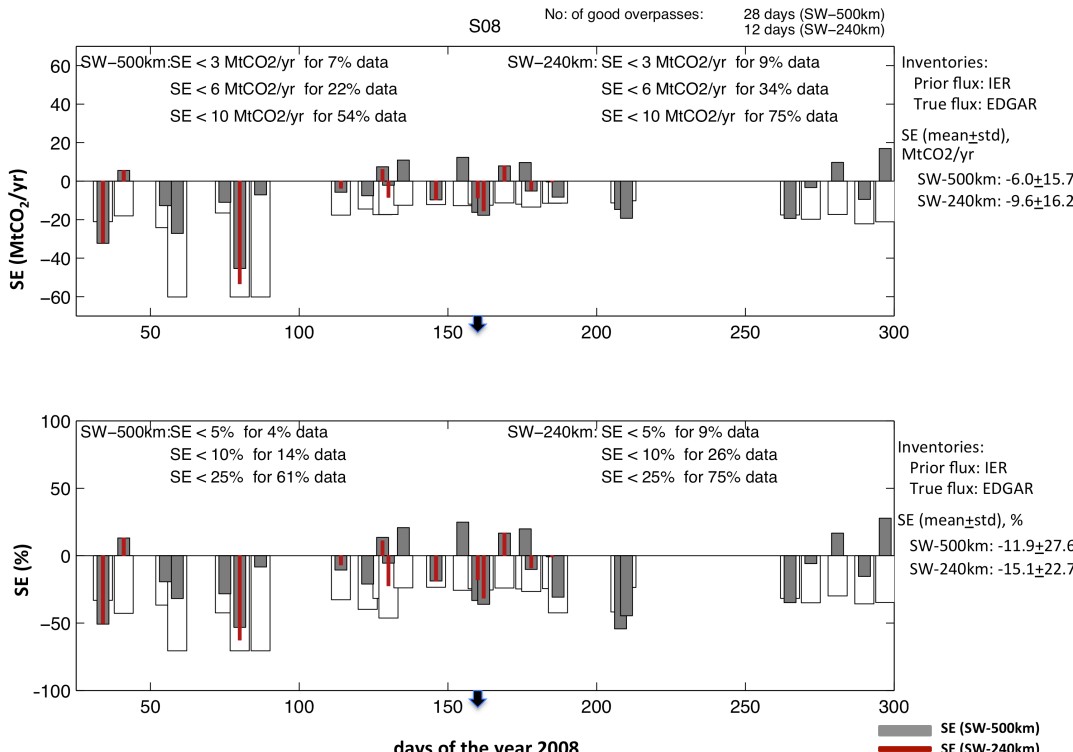

Figure 13: Similar to Fig. 10, but for the inversion experiment S08 using IER (a priori) and EDGAR (true) emission fluxes.



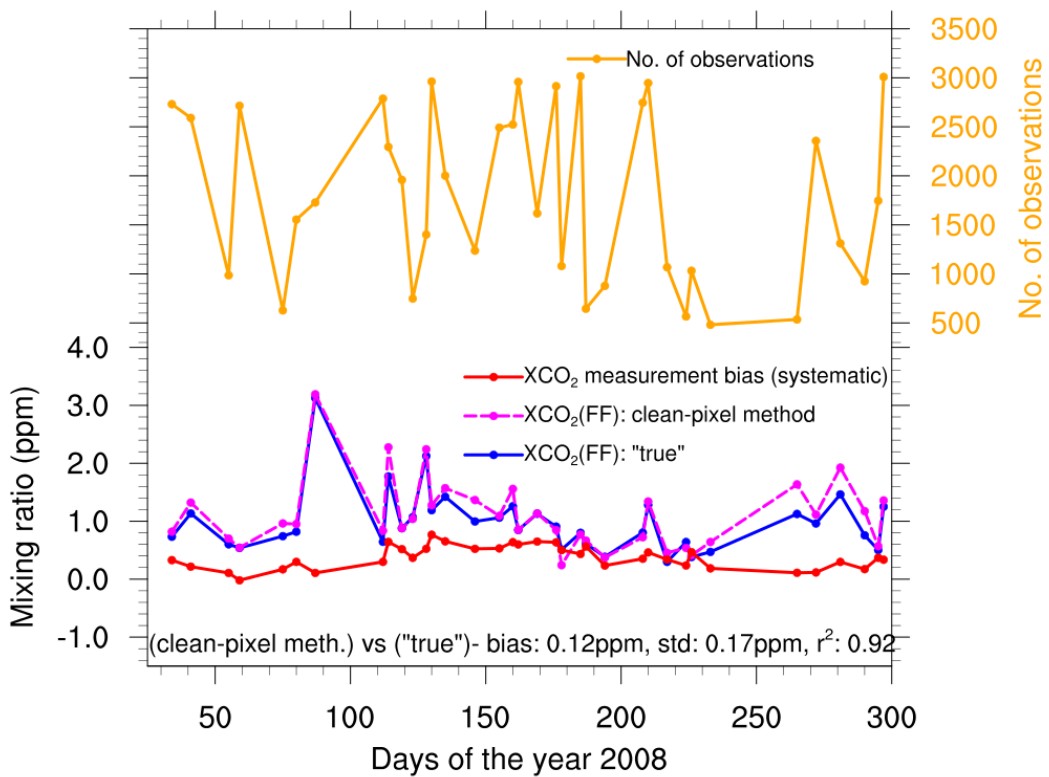

**Figure 14: Anthropogenic XCO₂ enhancements ("true" as well as those estimated by the clean-pixel method described in Sec. 5) and the CarbonSat's measurement biases (maximum values) over the target region around Berlin (100 km × 100 km) for all of CarbonSat's useful overpasses for the year 2008 (swath width: 500 km). The orange curve denotes the total number of CarbonSat simulated observations per overpass in the target region.**