# Peer review of "Tracking city CO2 emissions from space using a high resolution inverse modeling approach: A case study for Berlin, Germany"

_Atmospheric Chemistry and Physics, 2015_

## Referee Comment (RC1) · Anonymous Referee #1 · 5 Mar 2016

Review of "Tracking city emissions from space using a high resolution inverse modeling approach: A case study for Berlin, Germany" submitter for possible publication to ACP by D. Pillai et al.

This paper analyzes the potential of the ill-fated CarbonSat spaceborne radiometer to estimate CO2 emissions from a medium-sized city such as Berlin. "Ill-fated" is used above as the FLEX mission (vegetation fluorescence) was selected by ESA rather than CarbonSat for the forthcoming Earth-Explorer-8. Nevertheless, an instrument similar to CarbonSat may be proposed in the near future and the present study is therefore of interest to quantify the information content of such space observatory. The analysis is based on a classical Bayesian inversion that provides an estimate of the posterior

uncertainty. It makes use of an atmospheric transport model at 10 km resolution and a detailed evaluation of the random and systematic error of the observations. The impact of these errors on the emission estimates is quantified. In addition, the impact of other sources of error, such as the contribution from biospheric fluxes to the measured $CO_2$ and the imperfect knowledge of the spatial distribution of the emission are also evaluated. It is shown that the Berlin city emission can be evaluated with a relative uncertainty on the order of 20% and up to 30% when all sources of errors are combined. Note that the impact of atmospheric modeling error is not evaluated (and this is clearly stated in the paper). The paper presentation is very clear. It provides new results that can be of interest to a wide community. There is no doubt that this paper should eventually be published. I nevertheless would like to recommend a few changes (rather easy) that would, I think, improve the paper.

One criticism is that the accuracy needs in terms of city emissions are not discussed. In the introduction, it is explained that an independent evaluation of the emissions is needed to verify the impact of mitigation efforts. If this is the primary objective of the mission, I would thing that an accuracy of a few percent is needed. The introduction also mentions the objective of inventory verification. However, the accuracy that is required for such objective is not mentioned. Finally, the introduction mentions the potential of independent measurements of $CO_2$ for emission trading. However, I believe that emission trading is at the scale of industries, and not at the scale of the cities. I thus recommend a better description of the needs with a quantification of the accuracy requirement. Clearly, this question has a strong impact on the results. Indeed, the abstract conclude by saying that "CarbonSat is well suited to obtain city scale $CO_2$ emissions". Since there was no requirement set, one cannot draw such conclusion.

I have been confused with the size of the state vector $\lambda$. On the one hand, I understood (p7, line 20 and below) that only the scaling to prior emissions of the Berlin region was retrieved (the spatial and temporal variations of the emissions are assumed) so that there is a single element in the state vector. On the other hand, on page 8 (around line

5), it is stated that the prior uncertainty is a matrix with no correlation, which clearly indicates that there are several elements in the state vector. Please clarify.

Page 9 around line 30. I could not understand why a 500 km swath instrument leas to 25 valid observations during the year while the same with a swath width of 240 km leads to much more than half of this number. I would have expected that, as the swath is reduced by a factor of slightly more than 2, the number of valid observation be reduced by a factor of significantly more than 2. Please discuss. Is this number typical of what can expect for cities with similar cloud cover as Berlin, or is the CarbonSat orbit centered over Berlin which makes it a favourable case ?

P10 l-34. "A quite high scaling factor" (between CO2 and aerosol optical depth). Please explain the reasoning to state that it is a high scaling factor. Indeed, eventualy a CarbonSat-like mission may be used to monitor cities that are not as "clean" as Berlin. For such cities, the scaling factor chosen may be an underestimate.

I was surprised by the discussion of the "clean pixel method". In the present state, it is very hard to understand and it comes at od with the rest of the paper. I strongly suggest to remove this section.

P13 l 6. 14C is mentioned. I do not think that anyone believe that 14C can be measured from space. The present paper is about spaceborne observation and I think it is misleading to mention 14C here. As for the other tracers (CO, NOx), the authors do know that, when adding this source of information, one also adds an unknown variable (relative fraction of emissions). Thus, I think it is misleading to suggest that the concomitant measurement of these gases would allow a distinction of the biogenic and anthropogenic contributions.

The "conclusion section" is more a summary than a conclusion

P15 l 23. I could not understand the argument in the sentence "By showing that the systemic error of the retrieved fluxes. . .". Please rephrase

Finally, I have a recommendation for discussion : The simulations are made at 10 km resolution and the authors do not mention a significant loss of information from the original 2 km of the CarbonSat instrument. One should then wonder what is the added value of the high spatial resolution of CarbonSat. It seems that 10 km resolution is good enough to observe the plume from the Berlin city

Figure 2 is mentioned but not discussed. It does not bring anything to the paper and I thus strongly suggest to remove it.

In Figure 3, it seems that the original data (10 km resolution) went through spatial smoothing. I would like to see the pixels. Also, a zoom over the Berlin region would be appropriate

Figure 4 : Please use Y axis that start at zero. The current presentation is somewhat misleading. The difference between IER and EDGAR are surprisingly large. I wonder whether there are arguments to favor one versus the other.

Figure 6: There are features I do not understand: Around day 75, two successive prior emission values show differences by a factor of two. Based on Figure 4, I cannot understand how the weekly, seasonal or daily cycles can explain a difference by a factor of two (assuming the observation is around 11 when there are little hourly variations). Please investigate

---

## Referee Comment (RC2) · Anonymous Referee #2 · 29 Mar 2016

Review of the manuscript "Tracking city CO2 emissions from space using a high resolution Inverse modeling approach: A case study for Berlin, Germany" by Pillai et al.

This study presents the performance assessment of the CarbonSat instrument (which mission unfortunately has not been selected by ESA for the Earth Explorer 8 opportunity) for detection and quantification of the CO2 and CH4 city-emissions globally. Although CarbonSat was not selected, there is a clear need for such a mission specially designed and optimized to getting high resolution images of CO2 and CH4 from emission hot spots. From this perspective the synthetic data experiment like presented in this study is essential for further missions.

The authors estimate the utility of such potential instrument to disentangle anthro-

pogenic from natural emissions of greenhouse gases given systematic and random measurement errors. This work continues the study of Buchwitz et al. (2013) but with more sophisticated setup. Using Bayesian inversion method the typical expected range of errors for anthropogenic emissions was derived under different scenarios and test cases.

The paper is well written, structured and I recommend the paper to be published at ACP subject to very small revisions listed below.

P3 L10: "The goal swath width is 500 km, but a smaller swath width will likely be implemented to limit cost (ESA, 2015). " Here and elsewhere in the text, check the consistency with the fact that CarbonSat was not selected.

P3 L17: "...Buchwitz et al. (2013a)... " I could not find Buchwitz et al. (2013b) in references (which is also cited further in the manuscript). I supposed you mean the paper: "Carbon Monitoring Satellite (CarbonSat): assessment of scattering related atmospheric CO2 and CH4 retrieval errors and first results on implications for inferring city CO2 emissions" Buchwitz et al. AMTD, 2013 Please, confirm.

P4 L10: "41 vertical levels" Please, indicate the model top at hPa

P4 L26: "An overview of the flux optimization is shown in Fig. 2." I think the reference to Pillai et. al. 2012 is enough. As well as for P11 L19-20, "As can be seen in Fig. 2". I suggest to remove Fig. 2

P5 L11: "Figure 4 shows..." I suggest to keep consistency between figures and use a) b) c) etc. for different panels throughout the paper.

As remark, I suggest to add more clarifications in sections 2.2 and 3.2. At this shape it's hard to get into details of the inversion system. It would be useful to add dimensions for every component of the system. For example the way of constructing Jacobian K, which is sensitivity to parameters lambda (F perturbed = Fdlamda?) showing the dimensions may improve the readability for the reader.

P6 L16: "Eq. (4)" -> "Eq. (6)"

P8 L8: "Any error correlations are neglected, hence Sprior is set to be a diagonal matrix" - is the measurement error covariance matrix also diagonal? If so, add few words about this assumption, especially for CarbonSat-like XCO2 observations.

P10 L6: "...typically differs... " - Typically for Berlin region or in general?

P10 L16-17: "In general, we find that the two different swath widths have a negligible impact on the daily SE of the retrieved emissions" - Please, rephrase this sentence as in conclusion section.

P12 L25-26: "Furthermore, the systematic errors of the retrieved emission fluxes for both swath widths are found to be lower than the systematic error of the prior fluxes (estimated based on "true" fluxes) except for a very few cases,..." Please, rephrase this sentence.

P12 L12: "...in the target region is notably different." - Here need to add ref. to the figure 6 in the end of the sentence. Otherwise this figure is not mentioned in the paper at all.

P15 L22-23: "By showing that the systemic error of the retrieved fluxes is lower than that of the prior fluxes (estimated based on true fluxes) in most of the cases," – please, consider to rephrase this sentence

Also, as a comment to section 4.3 I think there might be effect of ignoring transport model uncertainty giving less weight to the prior fluxes.

As for section 5 - Discussion, I agree with Referee #1 about introduction and discussion of the "clean pixel method" here. From my point of view it disturbs the logic of the paper and I suggest to remove this paragraph.

reference:

M. Buchwitz, M. Reuter, H. Bovensmann, D. Pillai, J. Heymann, O. Schneising, V.
* * *
Rozanov, T. Krings, J. P. Burrows, H. Boesch, C. Gerbig, Y. Meijer, and A. LÂÍoscher: Carbon Monitoring Satellite (CarbonSat): assessment of scattering related atmospheric $CO_2$ and $CH_4$ retrieval errors and first results on implications for inferring city $CO_2$ emissions. Atmos. Meas. Tech. Discuss., 6, 4769–4850, 2013, www.atmos-meas-tech-discuss.net/6/4769/2013/, doi:10.5194/amtd-6-4769-2013

---

## Author Response (AR1)

**Tracking city CO₂ emissions from space using a high resolution inverse modeling approach: A case study for Berlin, Germany**

Authors' responses to reviewers' comments:

We would like to thank both referees for their careful reviewing and constructive comments, recommendations, and suggestions. Our responses to these comments are as follows:

**[RC]:** Reviewer's comment
**[AR]:** Authors' response
**[ME]:** Manuscript edits & modification

**Reviewer #1**

**[RC]** "One criticism is that the accuracy needs in terms of city emissions are not discussed. In the introduction, it is explained that an independent evaluation of the emissions is needed to verify the impact of mitigation efforts. If this is the primary objective of the mission, I would thing that an accuracy of a few percent is needed. The introduction also mentions the objective of inventory verification. However, the accuracy that is required for such objective is not mentioned. Finally, the introduction mentions the potential of independent measurements of CO2 for emission trading. However, I believe that emission trading is at the scale of industries, and not at the scale of the cities. I thus recommend a better description of the needs with a quantification of the accuracy requirement. Clearly, this question has a strong impact on the results. Indeed, the abstract conclude by saying that "CarbonSat is well suited to obtain city scale CO2 emissions". Since there was no requirement set, one cannot draw such conclusion. "

**[AR]** We agree that the paper can be improved in this respect and we will modify the Introduction and the Abstract to consider this comment. The text will be modified as follows:

> **[ME]** P2L13 in ACPD: "The reporting of the emissions of $CO_2$ is currently determined by national and regional agreements and legislation. This is an evolving topic for policy makers. For example, there exists an emission inventory which accounts for total annual U.S. emissions between 1990 and 2014 (EPA, 2016). In the European Union, the monitoring and reporting of greenhouse gas emissions are performed and regulated under the Commission Regulation (EU) No 601/2012 (European Commission, 2012). Similarly, the UK Government has announced, under the Companies Act 2006 (Strategic Report and Directors' Report) Regulations 2013, that companies are required to report their annual greenhouse gas emissions in their directors' report (see http://www.legislation.gov.uk/uksi/2013/1970/pdfs/uksi_20131970_en.pdf). There is also a guideline for national greenhouse inventories prepared by a task force of the IPCC (IPCC 2006). Following the agreement of the UNFCCC COP21 in Paris 2015, it is likely that new guidelines for reporting the emissions of greenhouse gases will be required.
>
> The uncertainties, i.e. the sum of systematic and stochastic error, in the national average of annual fossil fuel $CO_2$ emissions from the United states is estimated to be 2 to 5 % (EPA, 2016). The corresponding values for countries without well-developed energy sector statistics are even higher, giving rise to uncertainties of about 10 to 20 % at the national level (Gregg et al., 2008). When disaggregating these national emissions at fine scales (e.g. city scale) based

on conventional accounting methods, the associated uncertainties are expected to be significantly higher compared to those of national averages (Oda and Maksyutov, 2011). Hence reliable emission estimates are not often available at a scale relevant for urban emissions and the associated uncertainties. "

P2L20 in ACPD: "In order to assess accurately the contribution of a city or other emission hot spot to the $CO_2$ or other GHG emission, accurate knowledge of the surface fluxes at high spatial and temporal resolutions are needed. Ideally the accuracy of the estimated flux needs to be high for unambiguous attribution of source strength. The uncertainty of these estimations is required to be reduced to the extent that is feasible. In ESA (2015) it is noted (see their Sect. 4.1.2) that accuracies better than 10% would be useful for providing important additional information for cities where inventories exist, and accuracies better than 20% would contribute knowledge for cities where inventories do not exist."

"The results show that these potential space-based top-down flux estimates have high accuracy; hence this study contributes to the definition of achievable targets for emission fluxes at the city scale."

Abstract
P1L33 in ACPD: "Overall, we conclude that a satellite mission such as CarbonSat has high potential to obtain city-scale $CO_2$ emissions as needed to enhance our current understanding of anthropogenic carbon fluxes, and that CarbonSat-like satellites should be an important component of a future global carbon emission monitoring system."

**[RC]** I have been confused with the size of the state vector λ. On the one hand, I understood (p7, line 20 and below) that only the scaling to prior emissions of the Berlin region was retrieved (the spatial and temporal variations of the emissions are assumed) so that there is a single element in the state vector. On the other hand, on page 8 (around 5), it is stated that the prior uncertainty is a matrix with no correlation, which clearly indicates that there are several elements in the state vector. Please clarify.

**[AR]** The text will be modified to clarify this.

> **[ME]** P7L21 in ACPD: "The other element is a constant, i.e. $\lambda_0 = 0$, for the entire scene per overpass to account for variations of the background $XCO_2$ (see Eq. 4) and to treat the background variations independently of the city emissions as done in Buchwitz et al. (2013b)."

**[RC]** Page 9 around line 30. I could not understand why a 500 km swath instrument leas to 25 valid observations during the year while the same with a swath width of 240 km leads to much more than half of this number. I would have expected that, as the swath is reduced by a factor of slightly more than 2, the number of valid observation be reduced by a factor of significantly more than 2. Please discuss. Is this number typical of what can expect for cities with similar cloud cover as Berlin, or is the CarbonSat orbit centered over Berlin which makes it a favourable case ?

**[AR]** It is mentioned in Page 7 line 16 in ACPD that there are 41 days of potential overpasses over Berlin for the 500 km swath width, based on the quality filtering scheme as described in Buchwitz et al. (2013a). However, as stated in page 9 line 27 in ACPD, we have applied an additional quality filter based on the posteriori random error of the retrieved emission, i.e., results are not shown for the days where retrieved emission random error exceeds 25%. This resulted in the number of valid days

reducing to 25 for the 500 km swath width. The additional quality filter did not reduce much the valid observations for the 240 km swath width. The text will be modified as follows:

> **[ME]** P9L30 in ACPD: "Applying this additional quality criterion has further reduced the number of potential overpasses for the 500 km swath width. "

**[AR]** This number ("N") is something typical we would expect for cities with similar cloud and SZA.

> **[ME]** P9L31 in ACPD "The value obtained here for "N" useful overpasses is expected to be typical for other cities with similar cloud coverage and latitude."

**[RC]** "A quite high scaling factor" (between CO2 and aerosol optical depth). Please explain the reasoning to state that it is a high scaling factor. Indeed, eventualy a CarbonSat-like mission may be used to monitor cities that are not as "clean" as Berlin. For such cities, the scaling factor chosen may be an underestimate.

Will be modified in the text as follows:

> **[ME]** P9L34 in ACPD "To quantify the urban aerosol enhancement over a region around two power plants in Germany and to study their impact on emission estimates, Krings et al. (2011) followed the above criteria and used an AOD scaling factor of 0.05 per 1% (4 ppm) of local $\Delta XCO_2$. As compared to their study, we have used a much higher scaling factor of 0.2, i.e., the AOD change, $\Delta AOD$ at 550 nm is 0.2 per 4 ppm of local anthropogenic $\Delta XCO_2$."

**[RC]** I was surprised by the discussion of the "clean pixel method". In the present state, it is very hard to understand and it comes at od with the rest of the paper. I strongly suggest to remove this section.

**[AR]** The discussion about the clear-pixel method has been removed.

**[RC]** 14C is mentioned. I do not think that anyone believe that 14C can be measured from space. The present paper is about spaceborne observation and I think it is misleading to mention 14C here. As for the other tracers (CO, NOx), the authors do know that, when adding this source of information, one also adds an unknown variable (relative fraction of emissions). Thus, I think it is misleading to suggest that the concomitant measurement of these gases would allow a distinction of the biogenic and anthropogenic contributions.

**[AR]** The mention of $\delta^{14}C$ has been removed as it cannot be measured from space with available technology. However, we disagree with the following statement about the additional tracers. There are a number of studies (including satellite-based) which showed the potential of using multiple species as tracers for anthropogenic $CO_2$ emissions from fossil fuel combustion. We see the importance of potential synergies to utilize these measurements to separate the anthropogenic and biospheric parts of the signals. The text will be modified to include the citations of these previous studies.

> **[ME]** P13L5 in ACPD "To assess the relative contribution of biogenic and anthropogenic sources, one can utilize additional co-emitted tracers such as CO and $NO_X$ (Newman et al. 2013; Silva et al., 2013; Berezin et al., 2013 and Reuter et al. 2014). In the time frame of a

potential CarbonSat mission, Sentinel-5 will be providing data on CO and tropospheric $NO_2$ (Ingmann et al., 2012), which when combined with CarbonSat data is expected to provide information for the attribution of air masses originating from fossil fuel combustion. Depending on the extent of the variability and the possible uncertainties, we can also rely on the biospheric and global model simulations to differentiate different source-sink contributions."

**[RC]** I could not understand the argument in the sentence "By showing that the systemic error of the retrieved fluxes. . .". Please rephrase

**[AR]** done

> **[ME]** P15L22 in ACPD "By showing that the systemic error of the retrieved fluxes is lower than the difference between the prior fluxes and the true fluxes in most cases, the results from the inversion experiment build confidence in our uncertainty estimations and ensure that the optimization is done correctly."

**[RC]** Finally, I have a recommendation for discussion: The simulations are made at 10 km resolution and the authors do not mention a significant loss of information from the original 2 km of the CarbonSat instrument. One should then wonder what is the added value of the high spatial resolution of CarbonSat. It seems that 10 km resolution is good enough to observe the plume from the Berlin city

**[AR]** The point is already discussed in the manuscript. Please see page 8, line 28 and page 9, line 14 in ACPD. However, in order to make this point into the Discussion Sect., we modified the manuscript as follows:

> **[ME]**   P13L43 in ACPD "When using observations at CarbonSat's 2 km spatial resolution, as mentioned in Sec. 4.1, it is likely that the magnitude and variability of local anthropogenic $XCO_2$ enhancement would be higher than our estimation that is based on simulations at 10 km spatial resolution. One of the main advantages of CarbonSat's resolution is its ability to provide a large number of cloud-free observations, and this study identified the potential observations over Berlin by utilizing CarbonSat's 2 km spatial resolution. "

**[RC]** Figure 2 is mentioned but not discussed. It does not bring anything to the paper and I thus strongly suggest to remove it.

**[AR]** done

**[RC]** In Figure 3, it seems that the original data (10 km resolution) went through spatial smoothing. I would like to see the pixels. Also, a zoom over the Berlin region would be appropriate

**[AR]** done

**[RC]** Figure 4 : Please use Y axis that start at zero. The current presentation is somewhat misleading. The difference between IER and EDGAR are surprisingly large. I wonder whether there are arguments to favor one versus the other.

**[AR]** The y-axis range has been modified. Our analysis shows large difference in emission intensities between IER and EDGAR over Berlin; however, it is difficult to

say which one is more appropriate than the other. We will add following line in revised version of the manuscript:

> **[ME]** P5L15 in ACPD "However, based on available sources of information it is difficult to conclude which inventory is more accurate."

**[RC]** Figure 6: There are features I do not understand: Around day 75, two successive prior emission values show differences by a factor of two. Based on Figure 4, I cannot understand how the weekly, seasonal or daily cycles can explain a difference by a factor of two (assuming the observation is around 11 when there are little hourly variations). Please investigate

**[AR]** Those two successive prior emissions (days 75 and 80) correspond to 15.03.2008 (Saturday) and 20.03.2008 (Thursday). Since emissions are generally low during weekend, we can expect an emission difference (about a factor of 1.3 to 1.5) between day 75 and 80. However, we found some issues in the plotting routine of the code used for creating this figure – (1) a shift in the weekdays was not applied to IER 2000 dataset to utilize it for the year 2008 and to preserve the temporal pattern differences between weekdays and weekends (please see Sec. 2.1.1), and (2) slightly different spatial area was used for calculating total EDGAR fluxes over the target region. Fig. 6 in ACPD (will be Fig. 5 in the revised version of the manuscript) will be modified. Although this plotting routine is independent of WRF simulations (it was done separately and had been taken care of) or inversion routines (note that scaling factors are used), this bug has very minor effect on our results of REs and SEs where prior fluxes are used for the estimation. Hence Fig. 7, Fig(s). 10 to 13 in ACPD (will be Fig. 6, Fig(s). 9 to 12 in the revised manuscript) and Table 1 will be modified. Revised figures are shown at the bottom of this document:

In addition to this, some more details will be given in the revised manuscript as follows:

**[ME]** P9L4 in ACPD: "Note that those fluxes "seen" by CarbonSat can vary significantly from one overpass to the next, as the temporal variations show a strong diurnal cycle (see also Fig. 3), and the time elapsed to transport the plume to where it is observed by CarbonSat changes with wind speed."

**Reviewer #2**

**[RC]** P3 L10: "The goal swath width is 500 km, but a smaller swath width will likely be implemented to limit cost (ESA, 2015). " Here and elsewhere in the text, check the consistency with the fact that CarbonSat was not selected.

**[AR]** Done. Will be modified as follows.

> **[ME]** P3L10 in ACPD: "The goal swath width for the proposed CarbonSat mission was 500 km, but smaller swath widths were also considered to limit cost (ESA, 2015)."

**[RC]** P3 L17: "...Buchwitz et al. (2013a)... " I could not find Buchwitz et al. (2013b) in references (which is also cited further in the manuscript). I supposed you mean the paper: "Carbon Monitoring Satellite (CarbonSat): assessment of scattering related atmospheric CO2 and CH4 retrieval errors and first results on implications for inferring city CO2 emissions" Buchwitz et al. AMTD, 2013 Please, confirm.

**[AR]** Thank you pointing this out. Buchwitz et al. (2013a) is the AMT paper and Buchwitz et al. (2013b) is the one which is given above. The reference section has been corrected accordingly.

**[RC]** P4 L10: "41 vertical levels" Please, indicate the model top at hPa

**[AR]** Done. Will be modified as follows.

> **[ME]** P4L10 in ACPD: "and the model top is 1.0 hPa"

**[RC]** P4 L26: "An overview of the flux optimization is shown in Fig. 2." I think the reference to Pillai et. al. 2012 is enough. As well as for P11 L19-20, "As can be seen in Fig. 2". I suggest to remove Fig. 2

**[AR]** Done.

**[RC]** P5 L11: "Figure 4 shows..." I suggest to keep consistency between figures and use a) b) c) etc. for different panels throughout the paper.

**[AR]** Done.

**[RC]** As remark, I suggest to add more clarifications in sections 2.2 and 3.2. At this shape it's hard to get into details of the inversion system. It would be useful to add dimensions for every component of the system…..

**[AR]** Done. Will be modified as follows.

> **[ME]** P7L26 in ACPD: "… and the dimension of K is $n \times m$ where $n$ corresponds to the numbers of elements in the state vector and $m$ is the number of $XCO_2$ observations."

**[RC]** P6 L16: "Eq. (4)" -> "Eq. (6)"

**[AR]** Done. Thank you.

**[RC]** P8 L8: "Any error correlations are neglected, hence Sprior is set to be a diagonal matrix" - is the measurement error covariance matrix also diagonal? If so, add few words about this assumption, especially for CarbonSat-like XCO2 observations.

**[AR]** Will be modified. Please see our response to Referee #1

**[RC]** P10 L6: "...typically differs... " - Typically for Berlin region or in general?

**[AR]** in general

**[RC]** P10 L16-17: "In general, we find that the two different swath widths have a negligible impact on the daily SE of the retrieved emissions" - Please, rephrase this sentence as in conclusion section.

**[AR]** Will be modified as follows:

> **[ME]** P10L17 in ACPD: "…although decreasing the swath width reduces the "N" useful

overpasses."

**[RC]** P12 L25-26: "Furthermore, the systematic errors of the retrieved emission fluxes for both swath widths are found to be lower than the systematic error of the prior fluxes (estimated based on "true" fluxes) except for a very few cases,..." Please, rephrase this sentence.

**[AR]** Will be modified as follows:

> **[ME]** P12L25 in ACPD: "… swath widths are found to be lower than the difference between the prior fluxes and the "true" fluxes except for a very few cases…"

**[RC]** P12 L12: "...in the target region is notably different." - Here need to add ref. to the figure 6 in the end of the sentence. Otherwise this figure is not mentioned in the paper at all.

**[AR]** Will be modified as follows: (please note that Fig. 6 will be Fig.5 in the revised version of the manuscript)

> **[ME]** P9L4 in ACPD: "Figure 5 shows an overview of prior fluxes used for these inversions."
> P12L12 in ACPD: "notably different (see Fig. 5)."

**[RC]** P15 L22-23: "By showing that the systemic error of the retrieved fluxes is lower than that of the prior fluxes (estimated based on true fluxes) in most of the cases," – please, consider to rephrase this sentence

**[AR]** Will be modified. Please see our response to Referee #1.

**[RC]** Also, as a comment to section 4.3 I think there might be effect of ignoring transport model uncertainty giving less weight to the prior fluxes.

**[AR]** Yes, we do agree that there will be an impact of transport uncertainty, but the scope of this study is to estimate the retrieved flux uncertainties that are caused only by CarbonSat's related errors. This is already mentioned in the manuscript. Please see page 14, line 1 in ACPD.

As for section 5 - Discussion, I agree with Referee #1 about introduction and discussion of the "clean pixel method" here. From my point of view it disturbs the logic of the paper and I suggest to remove this paragraph.

**[AR]** Removed. Please see our response to Referee #1.

**Revised Table 1**

[revised manuscript text omitted]

The reporting of the emissions of $CO_2$ is currently determined by national and regional agreements and legislation. This is an evolving topic for policy makers. For example, there exists an emission inventory which accounts for total annual U.S. emissions between 1990 and 2014 (EPA, 2016). In the European Union, the monitoring and reporting of greenhouse gas emissions are performed and regulated under the Commission Regulation (EU) No 601/2012 (European Commission, 2012). Similarly, the UK Government has announced, under the Companies Act 2006 (Strategic Report and Directors' Report) Regulations 2013, that companies are required to report their annual greenhouse gas emissions in their directors' report (see http://www.legislation.gov.uk/uksi/2013/1970/pdfs/uksi_20131970_en.pdf). There is also a guideline for national greenhouse inventories prepared by a task force of the IPCC (IPCC 2006). Following the agreement of the UNFCCC COP21 in Paris 2015, it is likely that new guidelines for reporting the emissions of greenhouse gases will be required.

The uncertainties, i.e. the sum of systematic and stochastic error, in the national average of annual fossil fuel $CO_2$ emissions from the United states is estimated to be 2 to 5 % (EPA, 2016). The corresponding values for countries without well-developed energy sector statistics are even higher, giving rise to uncertainties of about 10 to 20 % at the national level (Gregg et al., 2008). When disaggregating these national emissions at fine scales (e.g. city scale) based on conventional accounting methods, the associated uncertainties are expected to be significantly higher compared to those of national averages (Oda and Maksyutov, 2011). Hence reliable emission estimates are not often available at a scale relevant for urban emissions and the associated uncertainties. This is problematic in terms of judging the effectiveness of emission reduction schemes or designing new management strategies for emission trading. Furthermore, uncertainties in emission estimates impose important limitations on regional carbon budget estimations derived by most atmospheric inverse frameworks (top-down approach), in which anthropogenic emission fluxes are assumed to be well-known (Corbin et al., 2010; Göckede et al., 2010; Gurney et al., 2002, 2005).

In order to assess accurately the contribution of a city or other emission hot spot to the $CO_2$ or other GHG emission, accurate knowledge of the surface fluxes at high spatial and temporal resolutions are needed. Ideally the accuracy of the estimated flux needs to be high for unambiguous attribution of source strength. The uncertainty of these estimations is required to be reduced to the extent that is feasible. In ESA (2015) it is noted (see their Sect. 4.1.2) that accuracies better than 10% would be useful for providing important additional information for cities where inventories exist, and accuracies better than 20% would contribute knowledge for cities where inventories do not exist.

[revised manuscript text omitted]
, however this is not the case for some other cities in Europe. Based on available sources of information, it is difficult to conclude which inventory is more accurate. The seasonal variability exhibited by EDGAR Berlin emissions is substantially larger than that of the IER inventory. Larger emissions are seen in the EDGAR inventory in winter months, with values approximately a factor of 1.5 higher than those in summer months. This results from the increased demand of domestic heating in winter. In terms of the seasonal variability of the Berlin city emissions, the IER inventory shows a relatively small difference in winter-summer emission patterns (temporal) as compared to EDGAR, and shows overall larger emissions in winter. Both inventories show lower emissions during weekends, consistent with the reduced demand of transportation and power consumption. The hourly averaged Berlin emissions provided by both inventories display peak values during 7 to 9 am and 5 to 7 pm (local times), reflecting morning

and evening rush hours in terms of city traffic. Interestingly the IER Berlin emissions show "delayed" morning rush hours on weekends, with a maximum value around 11 am (local time).

The significant difference between these inventories in both temporal and spatial scales implies that our current knowledge of urban-scale emissions is inadequate, even for Central Europe, which is relatively well characterized in terms of emissions compared to many other parts of the world. Note that a part of these emission differences is likely due to the different data compilation years of the IER and EDGAR inventories. This "knowledge gap" is also important in inverse-modeling-based estimations of the source-sink distribution of $CO_2$, in which fossil fuel fluxes are generally assumed to be known. How critical the effect of this assumption is depends on the impact of these differences in emissions (emission uncertainties) on modeled atmospheric mixing ratios, as well as on the transport errors that are included in the model-data mismatch error in the inverse modeling framework. The impact of emission uncertainties is further discussed in Sec. 4.1.

**2.2. Inverse optimization technique**

The inverse optimization utilizes observational constraints to adjust a subset of parameters $\lambda$ out of model parameters $p$ in the surface flux model $f_m(p)$ in order to obtain a modeled concentration consistent with the observations. Hence the anthropogenic atmospheric concentration $c$ (column averaged dry air mole fraction) at different locations and times can be represented as:

$$c - c_{bg} = \mathbf{F} f_m(\lambda) + \varepsilon_{error} \tag{1}$$

Here, the matrix $\mathbf{F}$ links the atmospheric concentration to a vector $f_m(\lambda)$ whose dimension is equal to the total number of surface flux elements, multiplied by total time steps. The vector $c_{bg}$ is the background column averaged dry air mole fraction i.e. the concentration due to the advection of upstream tracer concentrations. For the inversion, $f_m(\lambda)$ is assumed to be linearly dependent on $\lambda$ and is expressed as:

$$f_m(\lambda) = \phi \lambda \tag{2}$$

where $\lambda$ represents a vector of daily scaling factors of surface fluxes, and $\phi$ represents the surface flux field over the model domain.

A linear model is obtained by combining Eq(s). 1 and 2:

$$y = \mathbf{K} \lambda + \varepsilon_{error} \tag{3}$$

where the measurement vector $y$ is given by

$$y = c - c_{bg} \tag{4}$$

and, $c_{bg}$ is obtained by linearizing the model with a reference state $\lambda_0 = 0$ (see Eq. 1).

The Jacobian matrix that represents the sensitivity of the observations $y$ to the state vector $\lambda$ is given by

$$\mathbf{K} = \mathbf{F} \phi \tag{5}$$

The state vector and the Jacobian matrix are further described in Sec. 3.2. *A priori* knowledge of the surface fluxes, $\lambda_{prior}$, along with their uncertainties is incorporated in the Bayesian formulation. The term, $\varepsilon_{error}$, is assumed to follow the Gaussian distribution described by the error covariance matrices of the measurements, $\mathbf{S}_e$ and the prior estimate, $\mathbf{S}_{prior}$. The posterior estimate of $\lambda$ is obtained by minimizing the cost function, $J$, which is given as:

$$J(\lambda) = (y - \mathbf{K}\lambda)^T \mathbf{S}_e^{-1}(y - \mathbf{K}\lambda) + (\lambda - \lambda_{prior})^T \mathbf{S}_{prior}^{-1}(\lambda - \lambda_{prior}) \tag{6}$$

Analytically solving for the minimum of Eq. (6) gives the optimal estimate of the state vector of the scaling factors $\hat{\lambda}$, as well as the associated error covariance matrix of $\hat{\lambda}$, termed as the posterior uncertainty, $S_{\hat{\lambda}}$. These are expressed as follows (Rodgers, 2000):

$$\hat{\lambda} = (K^T S_e^{-1} K + S_{prior}^{-1})^{-1}(K^T S_e^{-1} y + S_{prior}^{-1}\lambda_{prior}) \tag{7}$$

$$S_{\hat{\lambda}} = (K^T S_e^{-1} K + S_{prior}^{-1})^{-1} \tag{8}$$

**3. Bayesian Inversion of CarbonSat measurements**

**3.1. Pseudo observations**

The inversion utilizes a one year dataset of CarbonSat simulated observations at a spatial resolution of 2 km × 2 km, generated using the WRF-GHG forward model (10 km × 10 km) as described in Sect. 2.1 and CarbonSat's retrieval error (2 km × 2 km), estimated using an error parameterization scheme based on the measurement characteristics as described in Buchwitz et al. (2013a). The error parameterization scheme, described in detail in Buchwitz et al.(2013a), is based on six parameters consisting of solar zenith angle (SZA) and scattering-related parameters such as albedo in the near-infrared (NIR) and the first shortwave-infrared (SWIR-1) bands, cirrus optical depth (COD), cirrus top height (CTH), and aerosol optical depth (AOD) at 550 nm. We use the "Level 2 error dataset" (L2e files), described in Buchwitz et al. (2013a), that contains the random and systematic errors of CarbonSat's $XCO_2$ retrievals based on the error parameterization scheme. CarbonSat is assumed to follow an orbit similar to NASA's Terra satellite (www.nasa.gov/terra/), but with an equator crossing time of 11:30 a.m. Hence, for specifying the CarbonSat's geolocation, the L2e files utilize the geolocation provided in the Terra Level 1 dataset for the year 2008, but modified to consider the difference in equator crossing time. This dataset contains fields such as geodetic coordinates, ground elevation, and solar and satellite zenith angles etc., determined using the spacecraft attitude and orbit, a digital elevation model, and information derived from various other datasets such as the Filled Land Surface Albedo Product, generated from MOD43B3 (http://modis-atmos.gsfc.nasa.gov/ALBEDO/) at a spatial resolution of 1 minute (2 km at equator, and < 1 km at the poles), which is used to account for surface albedo. The cirrus parameters are represented using a spatiotemporally smoothed (8$^o$ × 8$^o$ and 3 months) dataset of COD and CTH, originally derived from CALIOP (Cloud-Aerosol LIdar with Orthogonal Polarization) onboard CALIPSO (Cloud-Aerosol Lidar and Infrared Pathfinder Satellite Observations, Winker et al., 2009). Global aerosol data products from the "GEMS project" (http://gems.ecmwf.int/) at a spatiotemporal resolution of 1.125$^o$ × 1.125$^o$ and 12 hourly are used to account for aerosols (AOD). This dataset is based on the assimilation of MODIS data and we use the AOD at 550 nm. As described in Buchwitz et al. (2013a), the L2e dataset only contains those Carbonsat simulated observations which are approximately cloud-free as determined using a cloud mask obtained from MODIS Terra (using the MODIS cloud cover data product (MOD35) at a spatial resolution of about 1 km × 1 km). As the remaining ground pixels may still suffer from cloud contamination (e.g., due to "too high" amounts of thin cirrus) or other disturbances, a quality filtering scheme is applied which is based on retrieved (e.g., COD and AOD) and known quantities (e.g., SZA). The quality filtering scheme is described in Buchwitz et al. (2013a) and we use here only those ground pixels which are considered "good" according to this scheme.

Initially, we have identified all the potentially useful Berlin overpasses, i.e., overpasses where at least some CarbonSat simulated observations are present over Berlin and surroundings for a given CarbonSat orbit. We found that the maximum number of observations is obtained during the summer months due to most favorable observation conditions (less clouds for extended time periods and regions, high SZA, etc.). In total, there are 41 days (orbits) of potentially useful overpasses over Berlin for the year 2008 for a swath width of 500 km. Note that the number of overpasses are smaller in the figures shown later. This is because of an additional quality filtering procedure applied after the inverse optimization that is based on retrieved random errors, as explained later.

**3.2. Definition of the state vector and Jacobian matrix**

In the present study, the state vector has two elements. The first element $\lambda$ (the scalable parameter of the emission flux) corresponds to the scaling factor of emission fluxes for a trimmed model domain, i.e., a region around Berlin (spatial extent: approximately 100 km × 100 km, hereafter referred to as the "target region" (TR). The other element is a constant, i.e. $\lambda_0 = 0$, for the entire scene per overpass to account for variations of the background $XCO_2$ (see Eq. 4) and to treat the background variations independently of the city emissions as done in Buchwitz et al. (2013b). The temporal resolution of $\lambda$ is set to be daily, assuming no spatial variations within the target region. The prior value of this scaling factor, $\lambda_{prior}$, is set to unity.

The Jacobian matrix $\mathbf{K}$ relates the measurement vector $\boldsymbol{y}$ to the state vector $\lambda$, and has elements that represent the response in mixing ratios to the emission fluxes (see Eq. 5) and the dimension of K is $n \times m$ where $n$ corresponds to the numbers of elements in the state vector and $m$ is the number of $XCO_2$ observations. Since we do not have an adjoint model, these sensitivity functions are derived by perturbing each element of the emission flux field $\boldsymbol{\phi}$ over the target region by small increment and applying the forward model (WRF-GHG) to obtain the resulting perturbed concentration field ($\mathbf{C} + \Delta\mathbf{C}$) over the target region. Hence, $\mathbf{K}$ is calculated as follows:

[revised manuscript text omitted]

Note that in the previous section we have used the CarbonSat systematic $XCO_2$ retrieval errors as provided by the error parameterization scheme described in Buchwitz et al. (2013a). However, as explained in Buchwitz et al. (2013b), this scheme may underestimate aerosol related biases if the spatially (not aggregated) high-resolution CarbonSat simulated observations are used for applications like the one used here. The reason is that aerosol-related retrieval biases have been computed using quite smooth model aerosol input data sets, which might not be sufficient to represent the aerosol plume over Berlin.

To consider this, an additional error term has been defined which is referred to as "high resolution aerosol error" in this manuscript. In this sub-section we present results for scenario S02, where the measurement error used for S01 described in the previous section has been replaced by the high-resolution aerosol error contribution to the systematic measurement error. We also present results for scenario S04, where the measurement error is the sum of the S01 and S02 errors.

The method of computing the "high resolution aerosol error" is described in detail in Buchwitz et al. (2013b). Here we describe it briefly as follows. A local AOD enhancement has been computed by scaling the observed anthropogenic $XCO_2$ spatial pattern, i.e., the AOD enhancement is assumed to be perfectly correlated with the $CO_2$ emission plume of interest (see Fig. 7b and Fig. 8b). To quantify the urban aerosol enhancement over a region around two power plants in Germany and to study their impact on emission estimates, Krings et al. (2011) followed the above criteria and used an AOD scaling factor of 0.05 per 1 (4 ppm) % of local $\Delta XCO2$. As compared to their study, we have used a much high scaling factor of 0.2, i.e., 
[revised manuscript text omitted]
 co-emitted tracers such as CO and $NO_X$ (Newman et al. 2013; Silva et al., 2013; Berezin et al., 2013 and Reuter et al. 2014). In the time frame of a potential CarbonSat mission, Sentinel-5 will be providing data on CO and tropospheric $NO_2$ (Ingmann et al., 2012), which when combined with CarbonSat data is expected to provide information for the attribution of air masses originating from fossil fuel combustion. Depending on the extent of the variability and the possible uncertainties, we can also rely on the biospheric and global model simulations to differentiate different source-sink contributions.

By assuming that the biospheric patterns are accurately modeled and that these biogenic signals can be subtracted from the measurement vector to isolate the anthropogenic contribution of $XCO_2$, our simple inversion system is constructed such that it takes into account the impact of CarbonSat sampling errors on the retrieved city emissions over Berlin. The applicability of our results to a scenario where these assumptions are not valid needs to be examined, but the current set-up is not well suited for this purpose since we have not taken into account additional

state vectors for biospheric contributions. On the other hand, the current setup allows us to investigate the extremely pessimistic scenario where we assume that we cannot model the biospheric contribution at all (see Sec. 4.2.3).

When using observations at CarbonSat's 2 km spatial resolution, as mentioned in Sec. 4.1, it is likely that the magnitude and variability of local anthropogenic $XCO_2$ enhancement would be higher than our estimation that is based on simulations at 10 km spatial resolution. One of the main advantages of CarbonSat's resolution is its ability to provide a large number of cloud-free observations and this study identified the potential observations over Berlin by utilizing CarbonSat's 2 km spatial resolution.

Although we utilize high-resolution forward simulations, at present our inversion system uses only one scaling factor for the entire target region for each useful overpass. This means that the current set-up cannot provide posterior estimates for each pixel or emission sector within the target region. In other words, the flexibility to capture the true spatial variation of fluxes is more limited in our simple inversion system than in pixel- or parameter-wise inversions. Using this simple inversion system may thus overestimate the retrieved flux uncertainty. While interpreting our results, one should keep in mind that we do not specify other important sources of errors in the inversion system such as transport error. As previously noted, the main focus of this study is to estimate the retrieved flux uncertainties that are caused only by CarbonSat's measurement errors. However, these transport related errors, which provide proper weight to the observations depending on the capability of the transport model, need to be taken into account when estimating the total flux uncertainty via inverse modeling.

**6.  Summary and Conclusion**

In the present study, we examine the potential of a satellite mission like CarbonSat for improving the current knowledge on the surface-atmosphere exchange of atmospheric $CO_2$. A significant contribution by the CarbonSat greenhouse gas (GHG) observations will be the ability to retrieve the emissions of localized (moderate to strong $CO_2$ and $CH_4$) emission sources such as cities, power plants, methane seeps, etc., as a result of its unique sampling capability at high spatial resolution (approximately 2 km × 2 km) with a good spatial coverage using a much wide swath. To demonstrate this, we have investigated the error on the retrieved fluxes using synthetic data which are similar to that expected from CarbonSat. We have simulated emissions from a medium-size city (in terms economic contribution and trade) and assessed the capability to retrieve anthropogenic emission fluxes for the city and its surrounding region (Berlin-centered target region investigated here: ~100 km × 100 km) from CarbonSat simulated observations. The results show that these potential space-based top-down flux estimates have high accuracy; hence this study contributes to the definition of achievable targets for emission fluxes at the city scale.

[revised manuscript text omitted]
/. We would also like to thank François-Marie Bréon for his helpful suggestions and careful review. This study has received funding from ESA (projects LOGOFLUX-I and LOGOFLUX-II) and the State and the University of Bremen.

[revised manuscript text omitted]

**Figure 6:** Precision (random errors (RE)) of the retrieved emission fluxes obtained by the inverse optimization using one year of CarbonSat simulated observations. Results of two different swath widths (SW) – 500 km (grey) and 240 km (red) – are shown. $S_{prior}$ values are indicated with magenta bordered bars for visualizing the reduction in uncertainty. The top and bottom panels show RE in $MtCO_2$ $yr^{-1}$ and in % respectively. An overview of the statistical distribution of RE, separately for 500 km (grey) and 240 km (red) swath widths, is given inside the panel. The overall mean ± standard deviation is given outside the respective panels. The lower and upper limits of the X-axis (days of the year) is restricted accordingly as there are no good CarbonSat simulated observation during winter months. The arrow marker in the X-axis indicates a particular day (24[th] June 2008) shown in Figs. 4, 7, and 8.

[Figure]

**Figure 7: XCO₂ systematic error over the target region on 24ᵗʰ June 2008, assuming a CarbonSat swath width of 500 km. The six scenarios (S01 to S06) are shown with a label inside the respective panel. For S01, S02 and S04, these errors are estimated using the error parameterization scheme of Buchwitz et al. (2013a). The other scenarios additionally utilize biogenic XCO₂ variability in the target region (simulated by WRF-GHG) to derive XCO₂ systematic errors. Note that different color scales are used for S02 and S03. All units are given in ppm.**

[Figure]

Figure 8: Observed anthropogenic $XCO_2$ enhancement over the target region during a CarbonSat overpass on 24[th] June 2008 (swath width: 500 km). Different panels show anthropogenic $XCO_2$ enhancement, while considering $XCO_2$ systematic errors for different scenarios as shown in Fig.8. The "True" $XCO_2$ (fossil fuel (FF)) enhancement (i.e. without any uncertainties) is given in the bottom panel (g) for comparison. Note that an offset, labeled inside each panel, is subtracted from the anthropogenic $XCO_2$ enhancement to better visualize the details (for the figure only). All units are ppm.

[Figure]

**Figure 9:** Systematic errors (SE) of the retrieved emission fluxes for S01, obtained by the inverse optimization using one year of CarbonSat simulated observations. Results of two different swath widths (SW) – 500 km (grey) and 240 km (red) – are shown. (a) and (b) show SE in $MtCO_2$ $yr^{-1}$ and in % respectively. An overview of the statistical distribution of SE, separately for 500 km (grey) and 240 km (red) swath widths, is given inside the panel. The overall mean +/- standard deviation is given outside the respective panels.

[Figure]

**Figure 10:** Same as Fig.9, but for S02, quantifying the impact of the worst-case assumption used for aerosol-related biases.

[Figure]

**Figure 11:** Same as Fig.9, but for S03, quantifying the impact of the worst-case modeling related errors by assuming that biogenic $XCO_2$ variations cannot be modeled at all.

[Figure]

**Figure 12:** Similar to Fig. 9, but for the inversion experiment S09 using IER (a priori) and EDGAR (true) emission fluxes.